# INVERSEBENCH: BENCHMARKING PLUG-AND-PLAY DIFFUSION PRIORS FOR INVERSE PROBLEMS IN PHYSICAL SCIENCES

**Hongkai Zheng**[1,*], **Wenda Chu**[1,*], **Bingliang Zhang**[1,*], **Zihui Wu**[1,*], **Austin Wang**[1],
**Berthy T. Feng**[1], **Caifeng Zou**[1], **Yu Sun**[2], **Nikola Kovachki**[3], **Zachary E. Ross**[1],
**Katherine L. Bouman**[1], **Yisong Yue**[1]
[1]California Institute of Technology, [2]Johns Hopkins University, [3]NVIDIA

## ABSTRACT

Plug-and-play diffusion priors (PnPDP) have emerged as a promising research direction for solving inverse problems. However, current studies primarily focus on natural image restoration, leaving the performance of these algorithms in scientific inverse problems largely unexplored. To address this gap, we introduce INVERSEBENCH, a framework that evaluates diffusion models across five distinct scientific inverse problems. These problems present unique structural challenges that differ from existing benchmarks, arising from critical scientific applications such as optical tomography, medical imaging, black hole imaging, seismology, and fluid dynamics. With INVERSEBENCH, we benchmark 14 inverse problem algorithms that use plug-and-play diffusion priors against strong, domain-specific baselines, offering valuable new insights into the strengths and weaknesses of existing algorithms. To facilitate further research and development, we open-source the codebase, along with datasets and pre-trained models, at https://devzhk.github.io/InverseBench/.

## 1 INTRODUCTION

Inverse problems are fundamental in many domains of science and engineering, where the goal is to infer the unknown source from indirect and noisy observations. Example domains include astronomy (Chael et al., 2019), geophysics (Virieux & Operto, 2009), optical microscopy (Choi et al., 2007), medical imaging (Lustig et al., 2007), fluid dynamics (Iglesias et al., 2013), among others. These inverse problems are often challenging due to their ill-posedness, complexity in the underlying physics, and unknown measurement noise.

The use of diffusion models (DMs) (Sohl-Dickstein et al., 2015; Dhariwal & Nichol, 2021) for solving inverse problems has become increasingly popular. One attractive approach is PnPDP methods that use the DM as a plug-and-play prior (Wang et al., 2022; Dou & Song, 2024), where the inference objective is decomposed into the prior (using a pre-trained diffusion model) and the likelihood of fitting the observations (using a suitable forward model). The advantage of this idea is twofold: (1) As a powerful class of generative models, DMs can efficiently encode the complex and high-dimensional prior distribution, which is essential to overcome ill-posedness. (2) As plug-and-play priors, DMs can accommodate different problems without any re-training by decoupling the prior and likelihood. However, current algorithms are primarily evaluated and compared on a fairly narrow set of image restoration tasks such as inpainting, super-resolution, and deblurring (Kadkhodaie & Simoncelli, 2021; Song et al., 2023a; Mardani et al., 2024). These problems differ greatly from those from science and engineering applications such as geophysics (Virieux & Operto, 2009), astronomy (Porth et al., 2019), oceanography (Carton & Giese, 2008), and many other fields, which have very different structural challenges arising from the underlying physics. It is unclear how much insight can be carried over from image restoration to scientific inverse problems.

In this paper, we introduce INVERSEBENCH, a comprehensive benchmarking framework designed to evaluate PnP diffusion prior approaches in a systematic and easily extensible manner. We curate

---

*These authors contributed equally to this work.

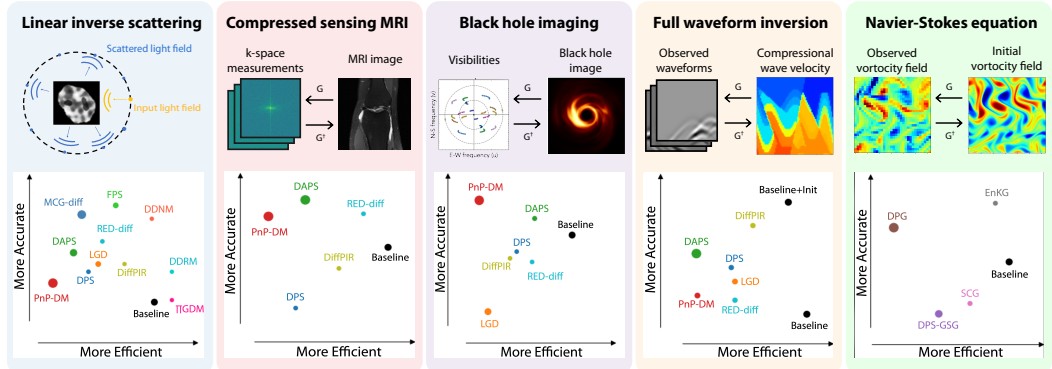

Figure 1: Illustration of five benchmark problems in the INVERSEBENCH. $G$ represents the forward model that produces observations from the source. $G^\dagger$ represents the inverse map. In the linear inverse scattering problem (left two), the observation is the recorded data from the receivers and the unknown source we aim to infer is the permittivity map of the object. The bottom panel displays the efficiency and accuracy plots for our benchmarked algorithms. Certain characteristics of the problem cause the efficiency and accuracy trade-offs of each algorithm to vary across tasks. In these plots, the larger radius of the points indicates greater interaction with the forward function $G$, as measured by the number of forward model evaluations.

a diverse set of five inverse problems from distinct scientific domains: optical tomography, black hole imaging, medical imaging, seismology, and fluid dynamics. These problems present structural challenges that differ significantly from natural image restoration tasks (cf. Figure 1 and Table 2), and encompass a broad spectrum of complexities across multiple scientific fields. Most notably, the forward model (which maps the source to observations) is defined using various types of physics-based models which can be highly nonlinear and difficult to evaluate.

We select 14 representative plug-and-play diffusion prior algorithms proposed for solving inverse problems, providing a thorough comparison of their performance across different scientific inverse problems and further insights into their efficacy and limitations. Additionally, we establish strong, domain-specific baselines for each inverse problem, providing a meaningful reference point for assessing the effectiveness of diffusion model-based approaches against traditional methods.

Through extensive experiments, we find that PnP diffusion prior methods generally exhibit strong performance given a suitable dataset for training a diffusion prior. This performance is consistent even as we vary the forward model (which is a strength of a PnP approach), given appropriate tuning. However, for forward models that require certain constraints on the input (e.g., uses a PDE solver), performance can be very sensitive to hyperparameter tuning. Moreover, the strength of using a diffusion prior can also be a limitation, as PnP diffusion prior methods have difficulty when the source image is out of the prior distribution (i.e., the use of diffusion models makes it difficult to recover "surprising" results). Additionally, we find that PnP methods that use multiple queries of the forward model tend to outperform simpler methods like DPS, at the cost of requiring additional tuning and computation, which points to an interesting direction for future method development.

INVERSEBENCH is implemented as a highly modular framework that can interface with new inverse problems and algorithms to run experiments at scale. We open-source the codebase, along with datasets and pre-trained models, at https://devzhk.github.io/InverseBench/.

## 2 PRELIMINARIES

### 2.1 INVERSE PROBLEMS

Following the typical setup, we have *observations* $\boldsymbol{y} \in \mathbb{C}^m$ from an unknown source $\boldsymbol{z} \in \mathbb{C}^n$ via a *forward model* $G : \mathbb{C}^n \to \mathbb{C}^m$. The inverse problem is to design a mapping $G^\dagger$ to infer $\boldsymbol{z}$ from $\boldsymbol{y}$:

$$\boldsymbol{z} \leftarrow G^\dagger(\boldsymbol{y}), \quad \text{where } \boldsymbol{y} = G(\boldsymbol{z}, \xi). \tag{1}$$

Here, $\xi$ represents noise in the forward model. In scientific applications, $G$ represents the measurement or sensing device (telescopes, infrared cameras, seismometers, electron microscopes, etc.). Inverse problems typically present four major challenges: (1) Many inverse problems are ill-posed, meaning that a solution may not exist, may not be unique, or may not be stable (Hadamard, 2014). For example, in black hole imaging, there could be multiple solutions that match the same sparse measurements. (2) The measurement noise is generally not separately observed (it is part of the observations $\boldsymbol{y}$), and accounting for it in the inverse problem can be challenging, especially for poorly characterized noise profiles (e.g., non-Gaussian). (3) The forward model might be highly nonlinear and lack a closed-form expression, leading to computational and numerical challenges in method design. (4) Designing an appropriate prior for the unknown source is also a critical challenge. For some problems, it is necessary for the designed prior to capture the complex structure of the solution space while remaining computationally tractable.

All these challenges necessitate some kind of regularization. While classic optimization approaches often employ simple regularizers (e.g., local isotropic smoothness), these fail to capture global or anisotropic properties. The use of diffusion models as a prior is attractive as a way to capture these more complex properties.

## 2.2 DIFFUSION MODELS

Diffusion models are a powerful class of deep generative models that can capture complicated high-dimensional distributions such as natural images (Rombach et al., 2022), proteins (Fu et al., 2024), small molecules (Luo et al., 2024), robotic trajectories (Chi et al., 2023), amongst other domains. Given their strong performance and compatibility with Bayesian inference, using diffusion models to model the solution space as a prior is a promising idea (Chung et al., 2023; Song et al., 2022).

We consider the continuous formulation of diffusion models proposed by Song et al. (2020), which expresses the forward diffusion and backward denoising process as stochastic differential equations (SDEs). The forward process transforms a data distribution $\boldsymbol{x}_0 \sim p_{\text{data}}$ into an approximately Gaussian one $\boldsymbol{x}_T \sim \mathcal{N}(0, \sigma^2(T)\boldsymbol{I})$ by gradually adding Gaussian noise according to:

$$\mathrm{d}\boldsymbol{x}_t = f(\boldsymbol{x}_t, t)\mathrm{d}t + g(t)\mathrm{d}\boldsymbol{w}_t, \tag{2}$$

where $f$ is a predefined vector-valued drift, $g$ is the diffusion coefficient, $\boldsymbol{w}$ is the standard Wiener process with time $t$ flowing from $0$ to $T$. The backward process sequentially denoises the Gaussian noise into clean data, which is given by the reverse-time SDE:

$$\mathrm{d}\boldsymbol{x}_t = \left( f(\boldsymbol{x}_t, t) - \frac{1}{2}g^2(t)\nabla_{\boldsymbol{x}_t} \log p_t(\boldsymbol{x}_t) \right) \mathrm{d}t + g(t)\mathrm{d}\bar{\boldsymbol{w}}_t, \tag{3}$$

where $p_t(\boldsymbol{x}_t)$ is the probability density of $\boldsymbol{x}_t$ at time $t$ and $\bar{\boldsymbol{w}}_t$ is the reverse-time Wiener process. The diffusion model is trained to learn the score function $\nabla_{\boldsymbol{x}_t} \log p_t(\boldsymbol{x}_t)$. Once trained, the diffusion model can generate new samples from the learned data distribution by solving Eq. (3).

## 2.3 PLUG-AND-PLAY DIFFUSION PRIORS FOR INVERSE PROBLEMS

We use the term *Plug-and-Play Diffusion Prior* (PnPDP) to refer to the class of recent methods that use diffusion models (or the denoising network within) as plug-and-play priors (Venkatakrishnan et al., 2013) for solving inverse problems. The basic idea is to either modify or use Eq. (3) to generate samples from $p(\boldsymbol{x}|\boldsymbol{y})$ rather than the prior $p(\boldsymbol{x})$, which under Bayes rule can be expressed as $p(\boldsymbol{x}|\boldsymbol{y}) \propto p(\boldsymbol{x})p(\boldsymbol{y}|\boldsymbol{x})$. The first term $p(\boldsymbol{x})$ can be modeled using a diffusion prior, and the second term $p(\boldsymbol{y}|\boldsymbol{x})$ can be computed using the forward model. Broadly speaking, existing PnPDP approaches can be grouped into four categories described below. Table 1 lists the 14 representative algorithms we selected, and notes their different requirements on the forward model. To avoid confusion, we use `Courier font` when referring to a specific algorithm in the main text throughout the paper (e.g., `PnP-DM` for Wu et al. (2024)).

**Guidance-based methods** Arguably the most popular approach to solving inverse problems with a pretrained diffusion model is guidance-based methods (Song et al., 2023a; Wang et al., 2022; Kawar et al., 2022; Rout et al., 2023; Chung et al., 2023), which modify Eq. (3) by adding a likelihood score term, $\nabla_{\boldsymbol{x}_t} \log p_t(\boldsymbol{y}|\boldsymbol{x}_t)$, along the diffusion trajectory. This term is related to the forward model $G$

Table 1: Requirements on the forward model of the algorithms evaluated in our experiments.

| Category | Method | SVD | Pseudo inverse | Linear | Gradient |
|---|---|---|---|---|---|
| Linear guidance | DDRM (Kawar et al., 2022) | ✓ | ✓ | ✓ | – |
| | DDNM (Wang et al., 2022) | ✗ | ✓ | ✓ | – |
| | ΠGDM (Song et al., 2023a) | ✗ | ✓ | ✗ | – |
| General guidance | DPS (Chung et al., 2023) | ✗ | ✗ | ✗ | ✓ |
| | LGD (Song et al., 2023b) | ✗ | ✗ | ✗ | ✓ |
| | DPG (Tang et al., 2023) | ✗ | ✗ | ✗ | ✗ |
| | SCG (Huang et al., 2024) | ✗ | ✗ | ✗ | ✗ |
| | EnKG (Zheng et al., 2024) | ✗ | ✗ | ✗ | ✗ |
| Variable-splitting | DiffPIR (Zhu et al., 2023) | ✗ | ✗ | ✗ | ✓ |
| | PnP-DM (Wu et al., 2024) | ✗ | ✗ | ✗ | ✓ |
| | DAPS (Zhang et al., 2024) | ✗ | ✗ | ✗ | ✓ |
| Variational Bayes | RED-diff (Mardani et al., 2023) | ✗ | ✗ | ✗ | ✓ |
| Sequential Monte Carlo | FPS (Dou & Song, 2024) | ✗ | ✗ | ✓ | – |
| | MCGDiff (Cardoso et al., 2024) | ✓ | ✓ | ✓ | – |

Table 2: Characteristics of different inverse problems in INVERSEBENCH, from left to right: whether the forward model is linear, whether one can compute the SVD from the forward model, whether the inverse problem operates in the complex domain, whether the forward model can be solved in closed form, whether one can access gradients from the forward model, and the noise type.

| Problem | Linear | SVD | Complex domain | Closed-form forward | Gradient access | Noise type |
|---|---|---|---|---|---|---|
| Linear inverse scattering | ✓ | ✓ | ✓ | ✓ | ✓ | Gaussian |
| Compressed sensing MRI | ✓ | ✗ | ✓ | ✓ | ✓ | Real-world |
| Black hole imaging | ✗ | ✗ | ✗ | ✓ | ✓ | Non-additive |
| Full waveform inversion | ✗ | ✗ | ✗ | ✗ | ✓ | Noise-free |
| Navier-Stokes equation | ✗ | ✗ | ✗ | ✗ | ✗ | Gaussian |

if the final clean $\boldsymbol{x}_0$ is a candidate source $\boldsymbol{z}$, in which case $p(\boldsymbol{y}|\boldsymbol{x}_0)$ can be estimated by querying $G$. However, $\log p_t(\boldsymbol{y}|\boldsymbol{x}_t)$ is generally intractable so various approximations have been proposed (Song et al., 2022; Chung et al., 2023; Song et al., 2023a; Boys et al., 2023).

**Variable splitting** Variable splitting is a widely used strategy for solving regularized optimization problems and conducting Bayesian inference (Vono et al., 2019; Chen et al., 2022; Lee et al., 2021). The core idea is to split the inference into two alternating steps (Wu et al., 2024; Zhu et al., 2023; Li et al., 2024a; Song et al., 2024; Zhang et al., 2024; Xu & Chi, 2024). The first step uses the forward model to update or sample in the neighborhood of the most recent $\boldsymbol{x}_t$. The second step runs unconditional inference on $p(\boldsymbol{x}_t)$, which amounts to running Eq. (3) for a small amount of time.

**Variational Bayes** Variational Bayes methods approximate intractable distributions such as $p(\boldsymbol{x}|\boldsymbol{y})$ using some simpler parameterized distribution $q_\theta$ (Zhang et al., 2018). The key idea is to find a $q_{\theta*}$ that, in a KL-divergence sense, both fits the observations $\boldsymbol{y}$ and agrees with the prior $p(\boldsymbol{x})$. Instead of directly sampling according to Eq. (3), it uses the diffusion model as a prior within a variational inference framework (Mardani et al., 2023; Feng et al., 2023; Feng & Bouman, 2024).

**Sequential Monte Carlo** Sequential Monte Carlo (SMC) methods draw samples iteratively from a sequence of probability distributions. These methods represent probability distributions by a set of particles with associated weights, which asymptotically converge to a target distribution following a sequence of proposal and reweighting steps. Recent works have extended SMC methods to the sequential diffusion sampling process (Wu et al., 2023; Trippe et al., 2023; Cardoso et al., 2024; Dou & Song, 2024), enabling zero-shot posterior sampling with diffusion priors. However, these methods are typically applicable only to inverse problems with linear forward models.

## 3 INVERSEBENCH

In this section, we introduce the formulation and specific challenges of the five scientific inverse problems considered in INVERSEBENCH: linear inverse scattering, compressed sensing MRI, black

hole imaging, full waveform inversion, and the Navier-Stokes equation. The characteristics of these inverse problems are summarized in Table 2. Their computational characteristics are summarized in Figure 6. Detailed descriptions and formal definitions can be found in Appendix B.

**Linear inverse scattering**  Inverse scattering is an inverse problem that arises from optical microscopy, where the goal is to recover the unknown permittivity contrast $\boldsymbol{z} \in \mathbb{R}^n$ from the measured scattered lightfield $\boldsymbol{y}_{\text{sc}} \in \mathbb{C}^m$. We consider the following formulation of inverse scattering

$$\boldsymbol{y}_{\text{sc}} = \boldsymbol{H}(\boldsymbol{u}_{\text{tot}} \odot \boldsymbol{z}) + \boldsymbol{n} \in \mathbb{C}^m \quad \text{where} \quad \boldsymbol{u}_{\text{tot}} = \boldsymbol{G}(\boldsymbol{u}_{\text{in}} \odot \boldsymbol{z}). \tag{4}$$

Here $\boldsymbol{G} \in \mathbb{C}^{n \times n}$ and $\boldsymbol{H} \in \mathbb{C}^{m \times n}$ are the discretized Green's functions that model the responses of the optical system, $\boldsymbol{u}_{\text{in}}$ and $\boldsymbol{u}_{\text{tot}}$ are the input and total lightfields, $\odot$ is the elementwise product, and $\boldsymbol{n}$ is the measurement noise. Since this problem is a linearized version of the general nonlinear inverse scattering problem based on the first Born approximation, we refer to it as linear inverse scattering. This problem allows us to test algorithms designed specifically for linear problems.

**Compressed sensing MRI**  Compressed sensing MRI is a technique that accelerates the scan time of MRI via subsampling. We consider the parallel imaging (PI) setup of CS-MRI, which is widely adopted in research and practice. Mathematically, PI CS-MRI can be formulated as an inverse problem that aims to recover an image $\boldsymbol{z} \in \mathbb{C}^n$ from

$$\boldsymbol{y}_j = \boldsymbol{P}\boldsymbol{F}\boldsymbol{S}_j\boldsymbol{z} + \boldsymbol{n}_j \in \mathbb{C}^m \quad \text{for } j = 1, ..., J$$

where $\boldsymbol{P} \in \{0, 1\}^{m \times n}$ is a subsampling operator and $\boldsymbol{F}$ is the Fourier transform; $\boldsymbol{y}_j$, $\boldsymbol{S}_j$, and $\boldsymbol{n}_j$ are the measurements, sensitivity map, and the noise of the $j$-th coil, respectively. Compressed sensing MRI is a linear problem, but it poses significant challenges due to its high-dimensional nature, involvement of priors in the complex domain, and attention to fine-grained details.

**Black hole imaging**  The measurements for black hole imaging (BHI) are obtained through Very Long Baseline Interferometry (VLBI). In this technique, each pair of telescopes $(a, b)$ provides a *visibility* (van Cittert, 1934; Zernike, 1938): a measurement that samples a particular spatial Fourier frequency of the source image related to the projected baseline between telescopes at time $t$

$$V_{a,b}^t = g_a^t g_b^t e^{-i\left(\phi_a^t - \phi_b^t\right)} \boldsymbol{I}_{a,b}^t(\boldsymbol{z}) + \boldsymbol{\eta}_{a,b}^t. \tag{5}$$

The ideal visibilities $\boldsymbol{I}_{a,b}^t(\boldsymbol{z})$, representing the Fourier component of the image $\boldsymbol{z}$, are corrupted by Gaussian thermal noise $\boldsymbol{\eta}_{a,b}^t$ as well as telescope-dependent amplitude errors $g_a^t$, $g_b^t$ and phase errors $\phi_a^t$, $\phi_b^t$ (EHTC, 2019a). To mitigate the impact of these amplitude and phase errors, derived data products called *closure quantities*, namely *closure phases* and *log closure amplitudes*, can be used to constrain inference (Blackburn et al., 2020):

$$\boldsymbol{y}_{t,(a,b,c)}^{\text{cp}} = \angle(V_{a,b}^t V_{b,c}^t V_{a,c}^t) \in \mathbb{R}, \quad \boldsymbol{y}_{t,(a,b,c,d)}^{\text{logca}} = \log\left(\frac{|V_{a,b}^t||V_{c,d}^t|}{|V_{a,c}^t||V_{b,d}^t|}\right) \in \mathbb{R}. \tag{6}$$

Here, $\angle$ and $|\cdot|$ denote the complex angle and amplitude. Given a total of $M$ telescopes, the number of closure phase measurements $\boldsymbol{y}_{t,(a,b,c)}^{\text{cp}}$ at time $t$ is $\frac{(M-1)(M-2)}{2}$, and the number of log closure amplitude measurements $\boldsymbol{y}_{t,(a,b,c,d)}^{\text{logca}}$ is $\frac{M(M-3)}{2}$, after accounting for redundancy. Closure quantities are nonlinear transformations of the visibilities, making a forward model that uses them for black hole imaging non-convex. The inverse problem is further complicated by the need for super-resolution imaging beyond the intrinsic resolution of the Event Horizon Telescope (EHT) observations (i.e., maximum probed spatial frequency), as well as phase ambiguities, which can lead to multiple modes in the posterior distribution (Sun & Bouman, 2021; Sun et al., 2024). Another challenge of BHI is that measurement noise is non-additive due to the usage of the closure quantities.

**Full waveform inversion**  Full waveform inversion (FWI) aims to infer subsurface physical properties (e.g. compressional and shear wave velocities) using the full information of recorded waveforms. In this work, we consider the problem of recovering the compressional wave velocity $v := v(\boldsymbol{x})$ (discretized as $\boldsymbol{z} \in \mathbb{R}^n$) from the observed wavefield $u_r$ (discretized as $\boldsymbol{y} \in \mathbb{R}^m$):

$$\boldsymbol{y} = \boldsymbol{P}\boldsymbol{u}, \tag{7}$$

where $\boldsymbol{P}$ is the sampling operator for receivers where observational data is available, and $\boldsymbol{u}$ is the discretization of the pressure wavefield $u := u(\boldsymbol{x}, t)$, which is a function of location $\boldsymbol{x}$ and time $t$. Here, $u$ is the solution to the acoustic (scalar) wave equation that models seismic wave propagation in heterogeneous acoustic media with constant density:

$$\frac{1}{v^2} \frac{\partial^2 u}{\partial t^2} - \nabla^2 u = q, \tag{8}$$

where $q := q(\boldsymbol{x}, t)$ is the source function (discretized as $\boldsymbol{q}$). Eq. (8) can be discretized as:

$$\boldsymbol{A}\boldsymbol{u} = \boldsymbol{q},$$

where $\boldsymbol{A}$ represents the discretized operator $\frac{1}{v^2} \frac{\partial^2}{\partial t^2} - \nabla^2$. Typically we only have observations at the free surface, the inverse problem has non-unique solutions. One of the major challenges for FWI is the prohibitive computational expense, especially for large problems, as it usually requires numerous calls to the forward modeling process. Moreover, the conventional method for FWI, called the adjoint-state method, casts it as a local optimization problem (Virieux et al., 2017; Virieux & Operto, 2009). This means that a sufficiently accurate initial model is required, as the solution is only sought in its vicinity. FWI conventionally needs to start with a smoothed model derived from simpler ray-based methods (Liu et al., 2017; Maguire et al., 2022), which imposes a significantly strong prior. A general method with less reliance on initialization is highly desired.

**Navier-Stokes equation** Navier-Stokes equation is a classic benchmarking problem from fluid dynamics (Iglesias et al., 2013). Its applications range from ocean dynamics to climate modeling where observations of the atmosphere are used to calibrate the initial condition for the downstream numerical forecasting. We consider the forward model that is given by the following 2D Navier-Stokes equation for a viscous, incompressible fluid in vorticity form on a torus.

$$\begin{aligned}
\partial_t \boldsymbol{w}(\boldsymbol{x}, t) + \boldsymbol{u}(\boldsymbol{x}, t) \cdot \nabla \boldsymbol{w}(\boldsymbol{x}, t) &= \nu \Delta \boldsymbol{w}(\boldsymbol{x}, t) + f(\boldsymbol{x}), & \boldsymbol{x} &\in (0, 2\pi)^2, t \in (0, T] \\
\nabla \cdot \boldsymbol{u}(\boldsymbol{x}, t) &= \boldsymbol{0}, & \boldsymbol{x} &\in (0, 2\pi)^2, t \in [0, T] \\
\boldsymbol{w}(\boldsymbol{x}, 0) &= \boldsymbol{w}_0(\boldsymbol{x}), & \boldsymbol{x} &\in (0, 2\pi)^2
\end{aligned} \tag{9}$$

where $\boldsymbol{u} \in C\left([0, T]; H_{\text{per}}^r((0, 2\pi)^2; \mathbb{R}^2)\right)$ for any $r > 0$ is the velocity field, $\boldsymbol{w} = \nabla \times \boldsymbol{u}$ is the vorticity, $\boldsymbol{w}_0 \in L_{\text{per}}^2\left((0, 2\pi)^2; \mathbb{R}\right)$ is the initial vorticity, $\nu \in \mathbb{R}_+$ is the viscosity coefficient, and $f \in L_{\text{per}}^2\left((0, 2\pi)^2; \mathbb{R}\right)$ is the forcing function. The solution operator $\mathcal{G}$ is defined as the operator mapping the vorticity from the initial vorticity to the vorticity at time $T$, i.e. $\mathcal{G} : \boldsymbol{w}_0 \to \boldsymbol{w}_T$. We consider the problem of recovering the initial vorticity field $\boldsymbol{z} := \boldsymbol{w}_0$ from the noisy partial observation $\boldsymbol{y}$ of the vorticity field $\boldsymbol{w}_T$ at time $T$ given by

$$\boldsymbol{y} = \boldsymbol{P}\boldsymbol{L}(\boldsymbol{z}) + \boldsymbol{n}$$

where $\boldsymbol{P}$ is the sampling operator, $\boldsymbol{n}$ is the measurement noise, and $\boldsymbol{L}(\cdot)$ is the discretized solution operator of Eq. (9). The Navier-Stokes equation does not admit a closed-form solution and thus there is no closed-form gradient available for the solution operator. Moreover, obtaining an accurate numerical gradient via automatic differentiation through the numerical solver is challenging due to the extensive computation graph expanded after thousands of discrete time steps.

## 4 EXPERIMENTS

### 4.1 EXPERIMENTAL SETUP

Here we provide a brief summary of our experimental setup. More details about the inverse problems and their corresponding datasets can be found in Appendix B. Technical details of DM pretraining can be found in Appendix B.6.

**Black hole imaging** We leverage a dataset of General Relativistic MagnetoHydroDynamic (GRMHD) (Mizuno, 2022) simulated black hole images as our training data. The training set consists of 50,000 resized 64×64 images. Since this dataset is not publicly available, we generate synthetic images from a pre-trained diffusion model for both the validation and test datasets. Specifically, we use 5 sampled images for the validation set and 100 sampled images for the test set.

**Full waveform inversion** We adapt the CurveFaultB dataset (Deng et al., 2022), which presents the velocity maps that contain faults caused by shifted rock layers. We resize the original data to resolution $128 \times 128$ with bilinear interpolation and anti-aliasing. The training set consists of 50,000 velocity maps. The test and validation sets contain 10 and 1 velocity maps, respectively.

**Linear inverse scattering** We create a dataset of fluorescence microscopy images using the online simulator (Wiesner et al., 2019). The training set consists of 10,000 HL60 nucleus permittivity images. The test and validation sets contain 100 and 10 permittivity images, respectively. We curate the test and validation samples so that all test samples have less than 0.6 cosine similarities to those in the training set.

**Compressed sensing MRI** We use the multi-coil raw $k$-space data from the fastMRI knee dataset (Zbontar et al., 2018). We exclude the first and last 5 slices of each volume for training and validation as they do not contain much anatomical information and resize all images down to $320 \times 320$ following the preprocessing procedure of (Jalal et al., 2021). In total, we use 25,012 images for training, 6 images for hyperparameter search, and 94 images for testing.

**Navier-Stokes** We create a dataset of non-trivial initial vorticity fields by first sampling from a Gaussian random field and then evolving Eq.9 for five time units. The equation setup follows Iglesias et al. (2013); Li et al. (2024b). We set the Reynolds number to 200 and spatial resolution to $128 \times 128$. The training set consists of 10,000 samples. The test and validation sets contain 10 and 1 samples, respectively.

**Pretraining of diffusion model priors** For each problem, we train a diffusion model on the training set using the pipeline from (Karras et al., 2022), and use the same checkpoint for all diffusion plug-and-play methods on each problem for a fair comparison. See more details in Appendix B.6.

## 4.2 EVALUATION METRICS

**Accuracy metrics** We use the Peak Signal-to-Noise Ratio (PSNR), Structure Similarity Index Measure (SSIM), as the generic ways to quantify recovery of the true source. For all the problems except for black hole imaging, we use the $\ell_2$ error $\|G(\hat{z}) - y\|_2$ to measure the consistency with the observation $y$. For black hole imaging, the closure quantities are invariant under translation, and so we measure the best fit under any shift alignment. We also assess the Blur PSNR, where images are blurred to match the target resolution of the telescope. We evaluate data misfit via the $\chi^2$ statistic on two closure quantities: the closure phase ($\chi^2_{\text{cp}}$) and the log closure amplitude ($\chi^2_{\text{logca}}$). A $\chi^2$ value close to 1 indicates better data fitting. To facilitate a comparison between underfitting ($\chi^2 > 1$) and overfitting ($\chi^2 < 1$), we report a unified metric defined as

$$\tilde{\chi}^2 = \chi^2 \cdot \mathbb{1}\{\chi^2 \geq 1\} + \frac{1}{\chi^2} \cdot \mathbb{1}\{\chi^2 < 1\}. \tag{10}$$

For FWI and Navier-Stokes experiments, we also use the relative $\ell_2$ error $\|\hat{z} - z\|_2 / \|z\|_2$ as it is a commonly used primary accuracy metric in PDE problems (Iglesias et al., 2013).

**Efficiency metrics** We define a set of efficiency metrics in Table 9 to evaluate the computational complexity of inverse algorithms more thoroughly. These metrics fall into two categories: (1) total metrics that measure the overall computational cost; (2) sequential metrics that help identify bottlenecks where forward model or diffusion model queries cannot be parallelized.

**Ranking score** To assess the relative ranking of different PnP diffusion models across various problems, we define the following ranking score for each problem. Given a set of accuracy or efficiency metrics $\{h_k\}_{k=1}^K$, we rank the algorithms according to each individual metric. Suppose algorithm $l$ has the rank $R_k(l)$ out of $L$ algorithms under the metric $k$. Its ranking score on this metric is given by $\text{score}_k(l) = 100 \times (L - R_k(l) + 1) / L$. For each problem, we calculate the average ranking score to assess overall performance:

$$\text{score}^{\text{problem}}(l) = \frac{1}{K} \sum_{k=1}^{K} \text{score}_k(l).$$

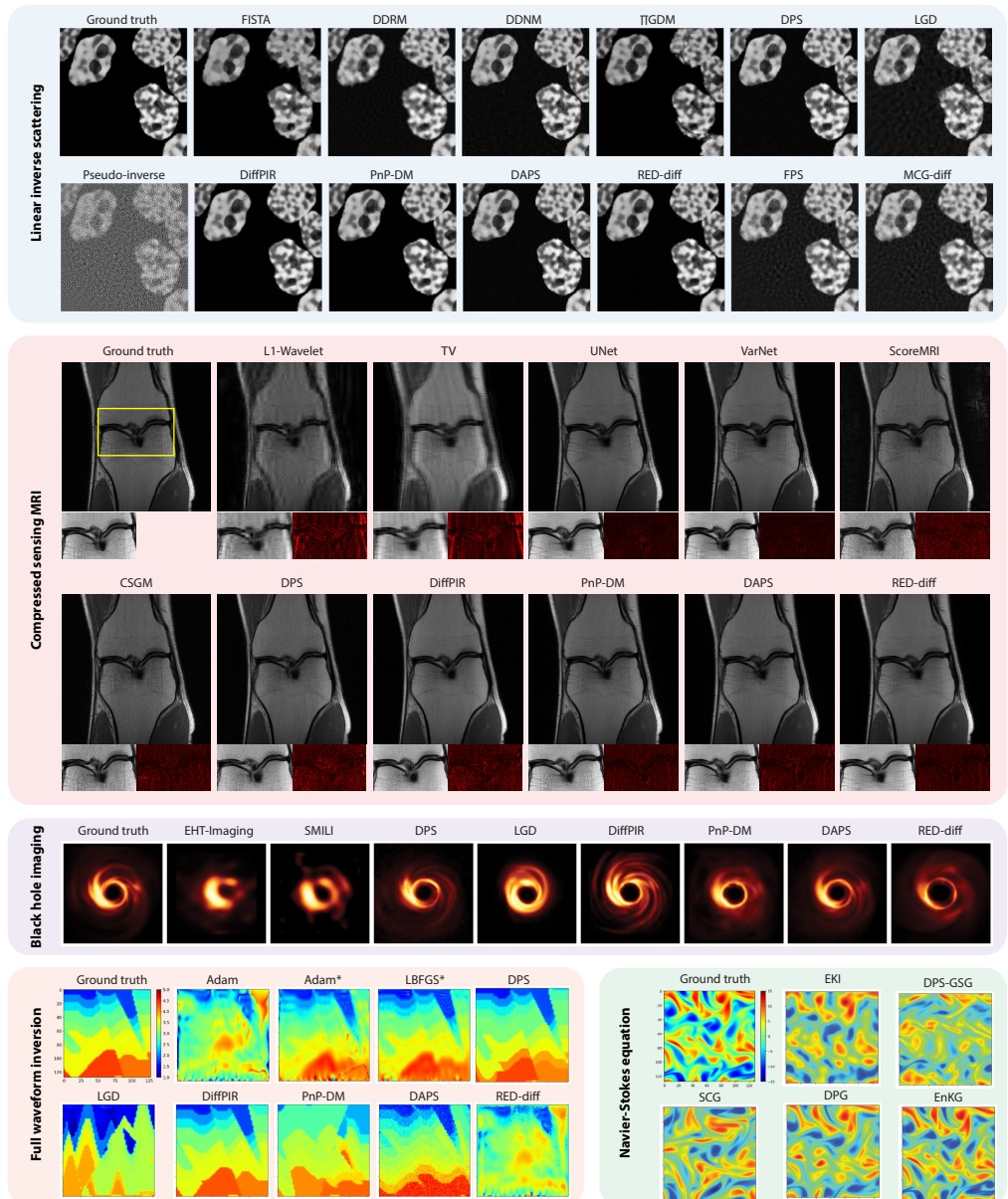

Figure 2: Qualitative comparison showing representative examples of PnP-DP methods and domain-specific baselines across five inverse problems. Note that for full waveform inversion, Adam* and LBFGS* are initialized with Gaussian-blurred ground truth, serving as references.

## 4.3 MAIN FINDINGS

The full experimental results for each problem are provided in Appendix A.1 as tables. Below, we highlight some key insights distilled from these results.

**How do PnPDP methods work compared to conventional baselines?** Our primary finding is that, given a suitable dataset for training a DM prior, PnPDP methods generally outperform conventional baselines. This is evident in Figure 1, where the PnPDP methods generally lie higher along the vertical axis. This finding is as expected given that the baselines do not incorporate such strong prior information.

However, if the classic optimization baselines are initialized well, then they sometimes outperform PnPDP methods, most of which cannot naturally incorporate an initialization beyond white noise. For example, in FWI, PnPDP methods clearly outperform the classic baseline methods if the baselines are initialized randomly or from a constant. But if initialized with a good guess (e.g., a heavily blurred ground truth image), they consistently outperform the current PnPDP methods. That being said, the fact that PnPDP methods rely much less on initialization than the traditional optimization methods is already an intriguing property. See qualitative comparison in Figure 1 and quantitative comparison in Table 7.

**How do PnPDP methods compare with each other?** In the problems where the forward model has a closed-form expression, methods that require more gradient queries, such as `DAPS` and `PnP-DM`, tend to be more accurate. However, since they have more queries to the forward model, they are also more expensive, as shown in Figure 1. Additionally, these methods require more careful tuning as they usually have larger hyperparameter spaces, as shown in Table 12[1].

In the problems where the forward model has no closed-form expression, particularly a forward model defined by a PDE system and implemented as a numerical PDE solver, this trend does not hold. In fact, `DAPS` and `PnP-DM` perform poorly, as shown in Figure 1 and Table 7. These methods also demonstrate an increased level of numerical instability and sensitivity to hyperparameters, as shown in Figure 3: minor adjustments in step size can lead to either unconditional generation results that ignore measurements (with slightly smaller steps) or complete failure (with slightly larger steps). This performance degradation stems from a critical limitation in many current PnPDP algorithms: they do not account for stability conditions required to query a forward model. For example, in the FWI and Navier-Stokes equation, the input of the forward model must satisfy the Courant–Friedrichs–Lewy (CFL) condition (Courant et al., 1967) to produce stable solutions. This issue is particularly pronounced for methods like `DAPS` and `PnP-DM`, which incorporate Langevin Monte Carlo (LMC) as an inner loop as LMC introduces additional Gaussian noise at each step, further exacerbating instability compared to other PnPDP methods.

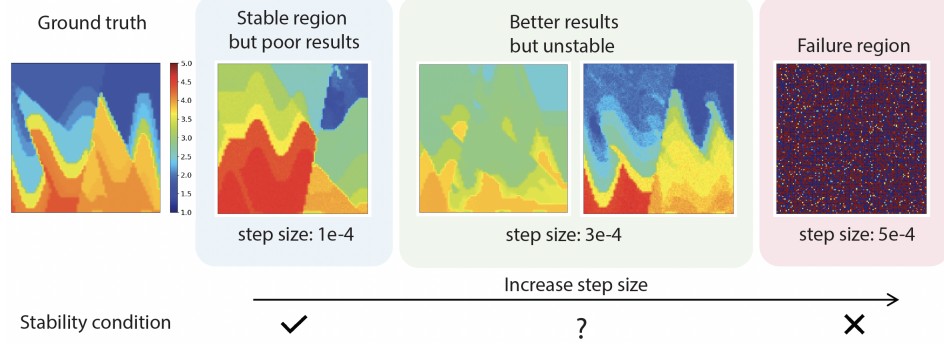

Figure 3: Illustration of the failures of PnPDP methods (`DAPS` as an example) on full waveform inversion. With a small learning rate, `DAPS` is numerically stable but does not solve the inverse problem effectively. With a slightly larger learning rate, `DAPS` produces a noisy velocity map that breaks the stability condition of the PDE solver, resulting in a complete failure.

**How does the performance vary with different levels of measurement sparsity?** As measurement sparsity increases, making the inverse problem more ill-posed, we observe an increasingly wide performance gap between PnPDP methods and baselines. Figure 4 illustrates this trend across three problems, showing that the average performance gain of top PnPDP methods over baselines grows with increasing measurement sparsity.

**How well do PnPDP methods deal with different forward models?** For linear inverse problems, our results demonstrate that PnPDP methods can effectively deal with varying forward models without the need for parameter tuning. To validate this, we conduct a controlled experiment in CS-MRI, where we maintain a consistent measurement sparsity while altering the subsampling pattern

---

[1]Note that tuning the hyperparameters of PnPDP approaches is still much more efficient than retraining a neural network that is typically required for end-to-end approaches.

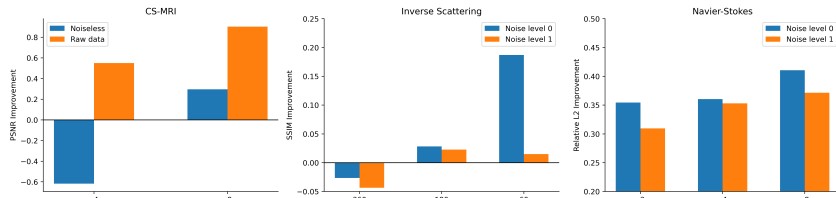

Figure 4: Relative performance of plug-and-play diffusion prior methods compared with traditional baselines under different levels of measurement sparsity on different tasks. Metrics are averaged over multiple PnPDP methods. The performance difference increases in general as the measurement becomes sparser.

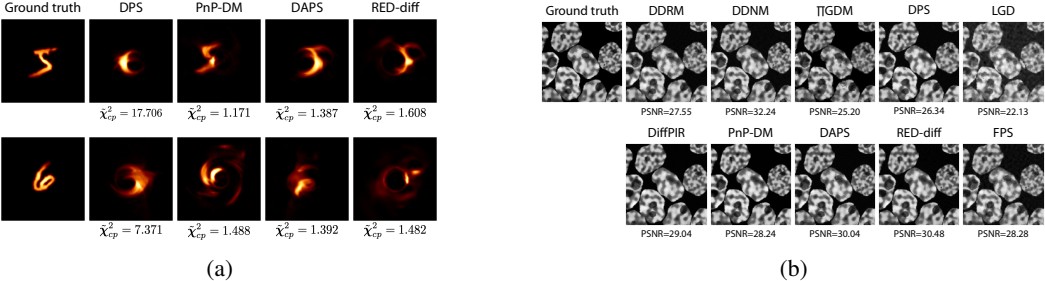

Figure 5: PnPDP methods on out-of-distribution test samples. (a) Black-hole imaging problem on digits inputs; and (b) inverse scattering on sources that contain 9 cells, while the prior model is trained on images with 1 to 6 cells.

(from vertical to horizontal lines). We assess the average performance variation across three method categories: traditional baselines, end-to-end approaches, and PnPDP methods. The average absolute performance change for PnPDP methods is 0.48dB (PSNR) and 0.016 (SSIM), comparable to the traditional baseline methods at 1.62dB (PSNR) and 0.027 (SSIM), but significantly smaller than the end-to-end methods, which exhibit changes of 9.58dB (PSNR) and 0.21 (SSIM). These findings indicate that PnPDP methods are more robust than both baseline and end-to-end methods when handling different forward models.[2]

**How well do PnPDP methods handle out-of-distribution sources?** In general, if the unknown source falls outside the diffusion prior distribution, PnPDP methods tend to generate solutions that are biased toward the prior. As illustrated in Figure 5a, most solutions produced by PnPDP methods exhibit a black hole ring feature characteristic of the diffusion prior. This suggests that while PnPDP approaches are flexible in capturing high-dimensional priors, they are limited in their ability to reliably recover "surprising" sources that lie outside the support of the diffusion prior distribution. However, when the unknown source is close to the diffusion prior distribution, PnPDP methods can recover it effectively, as demonstrated in Figure 5b.

## 5 DISCUSSION

We conclude by highlighting key research opportunities for advancing PnPDP methods in solving inverse problems. One research challenge we identify is that the current PnPDP methods do not account for stability conditions required to query a forward model, which leads to degraded performance and numerical instability. However, many scientific inverse problems are based on PDE systems that requires certain conditions on the inputs to simulate numerically and violating these constraints can result in meaningless solutions. Another direction for improvement is inference speed. As shown in Figure 1, almost all the PnPDP methods are less computationally efficient than the conventional baselines. There remains substantial room for optimization. Beyond these challenges, additional promising research directions such as robustness to model error and prior mismatch are further discussed in Appendix C.

---

[2] For end-to-end approaches, this is considered as an out-of-distribution test.

ACKNOWLEDGEMENT

This research is funded in part by NSF CPS Grant #1918655, NSF Award 2048237, NSF Award 2034306 and Amazon AI4Science Discovery Award. H.Z. is supported by the PIMCO and Amazon AI4science fellowship. Z.W. is supported by the Amazon AI4Science fellowship. B.Z. and W.C. are supported by the Kortschak Scholars Fellowship. B.F. is supported by the Pritzker Award and NSF Graduate Research Fellowship. Z.E.R. and C.Z. are supported by a Packard Fellowship from the David and Lucile Packard Foundation.

We thank Ben Prather, Abhishek Joshi, Vedant Dhruv, C.K. Chan, and Charles Gammie for the synthetic blackhole images GRMHD dataset used here, generated under NSF grant AST 20-34306.

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

# A APPENDIX

## A.1 TABLES OF MAIN RESULTS

Table 3: Results on Linear inverse scattering. PSNR and SSIM of different algorithms on linear inverse scattering. Noise level $\sigma_{\boldsymbol{y}} = 10^{-4}$.

| Number of receivers | **360** | | | **180** | | | **60** | | |
|---|---|---|---|---|---|---|---|---|---|
| Methods | PSNR | SSIM | Meas err (%) | PSNR | SSIM | Meas err (%) | PSNR | SSIM | Meas err (%) |
| **Traditional** | | | | | | | | | |
| FISTA-TV | 32.126 (2.139) | 0.979 (0.009) | 1.23 (0.25) | 26.523 (2.678) | 0.914 (0.040) | 2.65 (0.30) | 20.938 (2.513) | 0.709 (0.103) | 6.05 (0.65) |
| **PnP diffusion prior** | | | | | | | | | |
| DDRM | 32.598 (1.825) | 0.929 (0.012) | 1.04 (0.26) | 28.080 (1.516) | 0.890 (0.019) | 1.57 (0.39) | 20.436 (1.210) | 0.545 (0.037) | 3.04 (0.92) |
| DDNM | 36.381 (1.098) | 0.935 (0.017) | 0.78 (0.22) | 35.024 (0.993) | 0.895 (0.027) | 0.58 (0.16) | **29.235** (3.376) | 0.917 (0.022) | 0.28 (0.07) |
| ΠGDM | 27.925 (3.211) | 0.889 (0.072) | 2.74 (1.23) | 26.412 (3.430) | 0.816 (0.114) | 3.66 (1.79) | 20.074 (2.608) | 0.540 (0.198) | 6.90 (3.38) |
| DPS | 32.061 (2.163) | 0.846 (0.127) | 4.35 (1.19) | 31.798 (2.163) | 0.862 (0.123) | 4.28 (1.20) | 27.372 (3.415) | 0.813 (0.133) | 4.53 (1.31) |
| LGD | 27.901 (2.346) | 0.812 (0.037) | 1.17 (0.20) | 27.837 (2.337) | 0.803 (0.034) | 1.06 (0.16) | 20.491 (3.031) | 0.552 (0.077) | 1.45 (0.68) |
| DiffPIR | 34.241 (2.310) | **0.988** (0.006) | 1.11 (0.24) | 34.010 (2.269) | **0.987** (0.006) | 1.04 (0.23) | 26.321 (3.272) | 0.918 (0.028) | 1.27 (0.23) |
| PnP-DM | 33.914 (2.054) | **0.988** (0.006) | 1.21 (0.25) | 31.817 (2.073) | 0.981 (0.008) | 1.42 (0.26) | 24.715 (2.874) | 0.909 (0.046) | 2.20 (0.34) |
| DAPS | 34.641 (1.693) | 0.957 (0.006) | 1.03 (0.25) | 33.160 (1.704) | 0.944 (0.009) | 1.11 (0.25) | 25.875 (3.110) | 0.885 (0.030) | 1.51 (0.25) |
| RED-diff | **36.556** (2.292) | 0.981 (0.005) | 0.89 (0.23) | **35.411** (2.166) | 0.984 (0.004) | 0.87 (0.21) | 27.072 (3.330) | **0.935** (0.037) | 1.18 (0.23) |
| FPS | 33.242 (1.602) | 0.870 (0.026) | **0.70** (0.01) | 29.624 (1.651) | 0.710 (0.040) | **0.37** (0.01) | 21.323 (1.445) | 0.460 (0.030) | **0.15** (0.02) |
| MCG-diff | 30.937 (1.964) | 0.751 (0.029) | **0.70** (0.01) | 28.057 (1.672) | 0.631 (0.042) | 0.38 (0.01) | 21.004 (1.571) | 0.445 (0.028) | 0.21 (0.06) |

Table 4: Results on compressed sensing MRI. Mean and standard deviation are reported over 94 test cases. Underline: the best across all methods. Bold: the best across PnP DM methods.

| Subsampling ratio | **×4** | | | | | | **×8** | | | | | |
|---|---|---|---|---|---|---|---|---|---|---|---|---|
| Measurement type | Simulated (noiseless) | | | Raw | | | Simulated (noiseless) | | | Raw | | |
| Methods | PSNR↑ | SSIM ↑ | Data misfit ↓ | PSNR ↑ | SSIM ↑ | Data misfit ↓ | PSNR ↑ | SSIM ↑ | Data-fit | PSNR ↑ | SSIM ↑ | Data misfit ↓ |
| **Traditional** | | | | | | | | | | | | |
| Wavelet+$\ell_1$ | 29.45 (1.776) | 0.690 (0.121) | 0.306 (0.049) | 26.47 (1.508) | 0.598 (0.122) | 31.601 (15.286) | 25.97 (1.761) | 0.575 (0.105) | 0.318 (0.042) | 24.08 (1.602) | 0.511 (0.106) | 22.362 (10.733) |
| TV | 27.03 (1.635) | 0.518 (0.123) | 5.748 (1.283) | 26.22 (1.578) | 0.509 (0.123) | 32.269 (15.414) | 24.12 (1.900) | 0.432 (1.112) | 5.087 (1.049) | 23.70 (1.857) | 0.427 (0.112) | 23.048 (10.854) |
| **End-to-end** | | | | | | | | | | | | |
| Residual UNet | 32.27 (1.810) | 0.808 (0.080) | – | 31.70 (1.970) | 0.785 (0.095) | – | 29.75 (1.675) | 0.750 (0.088) | — | 29.36 (1.746) | 0.733 (0.100) | – |
| E2E-VarNet | 33.40 (2.097) | 0.836 (0.079) | – | 31.71 (2.540) | 0.756 (0.102) | – | 30.67 (1.761) | 0.769 (0.085) | — | 30.45 (1.940) | 0.736 (0.103) | – |
| **PnP diffusion prior** | | | | | | | | | | | | |
| CSGM | 28.78 (6.173) | 0.710 (0.147) | **1.518** (0.433) | 25.17 (6.246) | 0.582 (0.167) | 31.642 (15.382) | 26.15 (6.383) | 0.625 (0.158) | 1.142 (1.078) | 21.17 (8.314) | 0.425 (0.192) | **22.088** (10.740) |
| ScoreMRI | 25.97 (1.681) | 0.468 (0.087) | 10.828 (1.731) | 25.60 (1.618) | 0.463 (0.086) | 33.697 (15.209) | 25.01 (1.526) | 0.405 (0.079) | 8.360 (1.381) | 24.74 (1.481) | 0.403 (0.080) | 24.028 (10.663) |
| RED-diff | 29.36 (7.710) | 0.733 (0.131) | 0.509 (0.077) | **28.71** (2.755) | 0.626 (0.126) | 31.591 (15.368) | 26.76 (6.696) | 0.647 (0.124) | **0.485** (0.068) | **27.33** (2.441) | 0.563 (0.117) | 22.336 (10.838) |
| DiffPIR | 28.31 (1.598) | 0.632 (0.107) | 10.545 (2.466) | 27.60 (1.470) | 0.624 (0.111) | 34.015 (15.522) | 26.78 (1.556) | 0.588 (0.113) | 7.787 (1.741) | 26.26 (1.458) | 0.590 (0.113) | 24.208 (10.922) |
| DPS | 26.13 (4.247) | 0.620 (0.105) | 9.900 (2.925) | 25.83 (2.197) | 0.548 (0.116) | 35.095 (15.967) | 20.82 (4.777) | 0.536 (0.111) | 6.737 (1.928) | 23.00 (3.205) | 0.507 (0.109) | 24.842 (11.263) |
| DAPS | 31.48 (1.988) | 0.762 (0.089) | 1.566 (0.390) | 28.61 (2.197) | **0.689** (0.102) | **31.115** (15.497) | 29.01 (1.712) | 0.681 (0.098) | 1.280 (0.301) | 27.10 (2.034) | **0.629** (0.107) | 22.729 (10.926) |
| PnP-DM | **31.80** (3.473) | **0.780** (0.096) | 4.701 (0.675) | 27.62 (3.425) | 0.679 (0.117) | 32.261 (15.169) | **29.33** (3.081) | **0.704** (0.105) | 3.421 (0.504) | 25.28 (3.102) | 0.607 (0.117) | 22.879 (10.712) |

Table 5: Generalization results on compressed sensing MRI with ×4 acceleration and raw measurements. Mean and standard deviation are reported over 94 test cases.

| Generalization | Vertical → Horizontal | | | Knee → Brain | | | ×4 → ×8 | | |
|---|---|---|---|---|---|---|---|---|---|
| Methods | PSNR↑ | SSIM ↑ | Data misfit ↓ | PSNR ↑ | SSIM ↑ | Data misfit ↓ | PSNR ↑ | SSIM ↑ | Data misfit ↓ |
| **Traditional** | | | | | | | | | |
| Wavelet+$\ell_1$ | 27.75 (1.683) | 0.627 (0.133) | 31.744 (15.362) | 25.96 (1.253) | 0.747 (0.026) | 7.986 (0.965) | 24.08 (1.602) | 0.511 (0.106) | 22.362 (10.733) |
| TV | 28.18 (1.777) | 0.533 (0.138) | 32.311 (15.487) | 25.56 (1.302) | 0.686 (0.049) | 8.396 (0.990) | 23.70 (1.857) | 0.427 (0.112) | 23.048 (10.854) |
| **End-to-end** | | | | | | | | | |
| Residual UNet | 22.06 (1.682) | 0.603 (0.049) | – | 30.07 (1.364) | 0.881 (0.019) | – | 23.93 (2.176) | 0.610 (0.064) | – |
| E2E-VarNet | 22.13 (2.925) | 0.543 (0.103) | – | 31.97 (1.452) | 0.857 (0.038) | – | 24.59 (2.012) | 0.637 (0.069) | – |
| **PnP diffusion prior** | | | | | | | | | |
| CSGM | 26.56 (3.647) | 0.629 (0.129) | 31.866 (15.479) | 27.19 (7.521) | 0.779 (0.189) | 7.779 (1.043) | 21.17 (8.314) | 0.425 (0.192) | **22.088** (10.740) |
| ScoreMRI | 25.60 (1.647) | 0.473 (0.091) | 33.707 (15.274) | 28.52 (0.885) | 0.674 (0.045) | 9.472 (0.948) | 24.74 (1.481) | 0.403 (0.080) | 24.028 (10.663) |
| RED-diff | **28.95** (2.480) | 0.628 (0.126) | **31.740** (15.421) | **30.61** (0.982) | 0.811 (0.048) | **7.750** (0.996) | **27.33** (2.441) | 0.563 (0.117) | 22.336 (10.838) |
| DiffPIR | 27.93 (1.502) | 0.637 (0.113) | 34.188 (15.479) | 27.75 (0.854) | 0.823 (0.026) | 10.972 (1.016) | 26.26 (1.458) | 0.590 (0.113) | 24.208 (10.922) |
| DPS | 26.77 (1.546) | 0.571 (0.117) | 35.233 (16.006) | 26.77 (1.137) | 0.738 (0.031) | 10.806 (1.159) | 23.00 (3.205) | 0.507 (0.109) | 24.842 (11.263) |
| DAPS | 28.78 (2.209) | **0.696** (0.105) | 32.198 (15.538) | 29.29 (0.911) | 0.882 (0.025) | 8.255 (0.986) | 27.10 (2.034) | **0.629** (0.107) | 22.729 (10.926) |
| PnP-DM | 27.93 (3.444) | 0.689 (0.121) | 32.391 (15.235) | 29.96 (0.984) | **0.882** (0.028) | 8.789 (0.978) | 25.28 (3.102) | 0.607 (0.117) | 22.879 (10.712) |

Table 6: Results on black hole imaging. PSNR and Chi-squared of different algorithms on black hole imaging. Gain and phase noise and thermal noise are added based on EHT library.

| Observation time ratio | 3% | | | | 10% | | | | 100% | | | |
|---|---|---|---|---|---|---|---|---|---|---|---|---|
| Methods | PSNR | Blur PSNR | $\bar{\chi}^2_{cp}$ | $\bar{\chi}^2_{logca}$ | PSNR | Blur PSNR | $\bar{\chi}^2_{cp}$ | $\bar{\chi}^2_{logca}$ | PSNR | Blur PSNR | $\bar{\chi}^2_{cp}$ | $\bar{\chi}^2_{logca}$ |
| **Traditional** | | | | | | | | | | | | |
| SMILI | 18.51 (1.39) | 23.08 (2.12) | 1.478 (0.428) | 4.348 (3.827) | 20.85 (2.90) | 25.24 (3.86) | 1.209 (0.169) | 21.788 (12.491) | 22.67 (3.13) | 27.79 (4.02) | 1.878 (0.952) | 17.612 (10.299) |
| EHT-Imaging | 21.72 (3.39) | 25.66 (5.04) | **1.507** (0.485) | 1.695 (0.539) | 22.67 (3.46) | 26.66 (3.93) | **1.166** (0.156) | **1.240** (0.205) | 24.28 (3.63) | 28.57 (4.52) | **1.251** (0.250) | 1.259 (0.316) |
| **PnP diffusion prior** | | | | | | | | | | | | |
| DPS | 24.20 (3.72) | **30.83** (5.58) | 8.024 (24.336) | 5.007 (5.750) | 24.36 (3.72) | 30.79 (5.75) | 13.052 (43.087) | 6.614 (26.789) | 25.86 (3.90) | **32.94** (6.19) | 8.759 (37.784) | 5.456 (24.185) |
| LGD | 22.51 (3.76) | 28.50 (5.49) | 15.825 (16.838) | 12.862 (12.663) | 22.08 (3.75) | 27.48 (5.09) | 10.775 (21.684) | 13.375 (56.397) | 21.22 (3.64) | 26.06 (4.98) | 13.239 (17.231) | 13.233 (39.107) |
| RED-diff | 20.74 (2.62) | 26.10 (3.35) | 6.713 (6.925) | 9.128 (19.052) | 22.53 (3.02) | 27.67 (4.53) | 2.488 (2.925) | 4.916 (13.221) | 23.77 (4.13) | 29.13 (6.22) | 1.853 (0.938) | 2.050 (2.361) |
| PnPDM | **24.25** (3.45) | 30.49 (4.93) | 2.201 (1.352) | **1.668** (0.551) | **24.57** (3.47) | **30.80** (5.22) | 1.433 (0.417) | 1.336 (0.478) | **26.07** (3.70) | 32.88 (6.02) | 1.311 (0.195) | **1.199** (0.221) |
| DAPS | 23.54 (3.28) | 29.48 (4.88) | 3.647 (3.287) | 2.329 (1.354) | 23.99 (3.56) | 30.10 (5.13) | 1.545 (0.705) | 2.253 (9.903) | 25.60 (3.64) | 32.78 (5.68) | 1.300 (0.324) | 1.229 (0.532) |
| DiffPIR | 24.12 (3.25) | 30.45 (4.88) | 14.085 (14.105) | 10.545 (8.860) | 23.84 (3.59) | 30.04 (5.03) | 5.374 (3.733) | 5.205 (5.556) | 25.01 (4.64) | 31.86 (6.56) | 3.271 (1.623) | 2.970 (1.202) |

Table 7: Results on FWI. Mean and standard deviation are reported over 10 test cases. †: initialized from data blurred by Gaussian filters with $\sigma = 20$. ∗: one test case is excluded from the results due to numerical instability.

| Methods | Relative L2↓ | PSNR↑ | SSIM↑ | Data misfit↓ |
|---|---|---|---|---|
| **Traditional** | | | | |
| Adam | 0.333 (0.086) | 9.968 (2.083) | 0.305 (0.120) | 115.14 (52.10) |
| Adam† | 0.089 (0.021) | 21.273 (2.045) | 0.679 (0.073) | 15.89 (10.16) |
| LBFGS† | 0.070 (0.023) | 23.398 (2.749) | 0.704 (0.077) | 9.18 (6.47) |
| **PnP diffusion prior** | | | | |
| DPS | 0.250 (0.154) | 14.111 (6.820) | 0.491 (0.161) | 155.08 (92.17) |
| LGD | 0.244 (0.024) | 12.288 (0.889) | 0.341 (0.047) | 258.47 (26.40) |
| DiffPIR | 0.204 (0.129) | **16.113** (6.962) | **0.554** (0.191) | **88.53** (56.91) |
| DAPS† | **0.201** (0.103) | 14.914 (4.184) | 0.321 (0.067) | 111.13 (71.33) |
| PnP-DM | 0.259 (0.075) | 11.983 (2.269) | 0.431 (0.073) | 308.84 (26.34) |
| REDDiff | 0.319 (0.102) | 10.372 (2.650) | 0.280 (0.108) | 94.67 (41.33) |

Table 8: Results on Navier-Stokes equation. Relative $\ell_2$ error of different algorithms on 2D Navier-Stokes inverse problem, reported over 10 test cases. ∗: one or two test cases are excluded from the results due to numerical instability.

| Subsampling ratio | ×**2** | | | ×**4** | | | ×**8** | | |
|---|---|---|---|---|---|---|---|---|---|
| Measurement noise | $\sigma = 0.0$ | $\sigma = 1.0$ | $\sigma = 2.0$ | $\sigma = 0.0$ | $\sigma = 1.0$ | $\sigma = 2.0$ | $\sigma = 0.0$ | $\sigma = 1.0$ | $\sigma = 2.0$ |
| **Traditional** | | | | | | | | | |
| EKI | 0.577 (0.138) | 0.609 (0.119) | 0.673 (0.107) | 0.579 (0.145) | 0.669 (0.131) | 0.805 (0.112) | 0.852 (0.167) | 0.940 (0.115) | 1.116 (0.090) |
| **PnP diffusion prior** | | | | | | | | | |
| DPS-fGSG | 1.687 (0.156) | 1.612 (0.173) | 1.454 (0.154) | 1.203* (0.122) | 1.209* (0.116) | 1.200* (0.100) | 1.246* (0.108) | 1.221* (0.082) | 1.260 (0.117) |
| DPS-cGSG | 2.203* (0.314) | 2.117 (0.295) | 1.746 (0.191) | 1.175* (0.079) | 1.133* (0.095) | 1.114* (0.144) | 1.186* (0.117) | 1.204* (0.115) | 1.218 (0.113) |
| DPG | 0.325 (0.188) | 0.408* (0.173) | 0.466 (0.171) | 0.322 (0.200) | 0.361 (0.187) | **0.454** (0.207) | 0.596 (0.301) | 0.591 (0.262) | 0.846 (0.251) |
| SCG | 0.908 (0.600) | 0.928 (0.557) | 0.966 (0.546) | 0.869 (0.513) | 0.926 (0.546) | 0.929 (0.505) | 1.260 (0.135) | 1.284 (0.117) | 1.347 (0.141) |
| EnKG | **0.120** (0.085) | **0.191** (0.057) | **0.294** (0.061) | **0.115** (0.064) | **0.271** (0.053) | 0.522 (0.136) | **0.287** (0.273) | **0.546** (0.212) | **0.773** (0.170) |

Table 9: Table of metrics we use to capture the computation complexity of each algorithm.

| Metric | Description |
|---|---|
| # Fwd$_{total}$ | total forward model evaluations |
| # DM$_{total}$ | total diffusion model evaluations |
| # Fwd Grad$_{total}$ | total forward model gradient evaluations |
| # DM Grad$_{total}$ | total diffusion model gradient evaluations |
| Cost$_{total}$ | total runtime |
| # Fwd$_{seq}$ | sequential forward model evaluations |
| # DM$_{seq}$ | sequential diffusion model evaluations |
| # Fwd Grad$_{seq}$ | sequential forward model gradient evaluations |
| # DM Grad$_{seq}$ | sequential diffusion model gradient evaluations |
| Cost$_{seq}$ | sequential runtime |

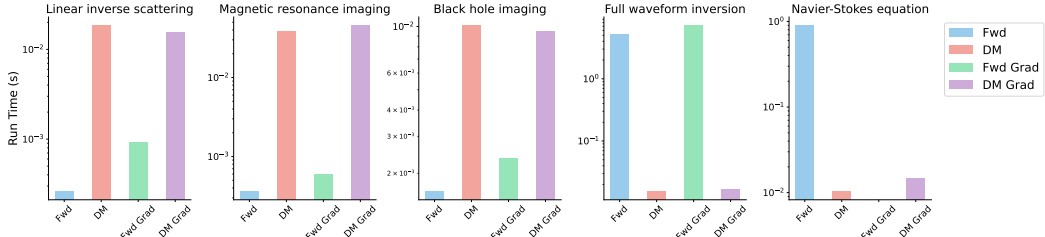

Figure 6: Computational characteristics of each forward model. Fwd: runtime of a single forward model evaluation tested on a single A100 GPU. DM: runtime of a single diffusion model evaluation. Fwd Grad: runtime of a single forward model gradient evaluation. DM Grad: runtime of a single diffusion model gradient evaluation. Note that the inverse problem of the Navier-Stokes equation only permits black-box access to the forward model so its Fwd Grad has no value.

## A.2 EXTENDED EVALUATION OF CS-MRI

Table 10: Diagnostic performance of compressed sensing MRI reconstructions.

| Method | Precision | Recall | mAP50 | mAP50 ranking | PSNR | SSIM | Data misfit | PSNR ranking |
|---|---|---|---|---|---|---|---|---|
| **Traditional** | | | | | | | | |
| Wavelet+$\ell_1$ | 0.532 | 0.332 | 0.385 | 9 | 28.16 (1.724) | 0.685 (0.064) | 23.501 (10.475) | 8 |
| TV | 0.447 | 0.251 | 0.263 | 11 | 28.31 (1.834) | 0.662 (0.079) | 24.182 (10.613) | 7 |
| **End-to-End** | | | | | | | | |
| Residual UNet | 0.482 | 0.462 | 0.439 | 8 | 31.62 (1.635) | 0.803 (0.050) | – | 2 |
| E2E-VarNet | **0.610** | 0.514 | **0.500** | **1** | **32.25** (1.901) | **0.805** (0.056) | – | **1** |
| **PnP diffusion prior** | | | | | | | | |
| CSGM | 0.501 | 0.528 | 0.454 | 6 | 27.34 (2.770) | 0.673 (0.082) | 23.483 (10.651) | 9 |
| ScoreMRI | 0.412 | 0.554 | 0.470 | 5 | 26.86 (2.583) | 0.547 (0.092) | 25.677 (10.491) | 10 |
| RED-diff | 0.478 | 0.468 | 0.448 | 7 | 31.56 (2.337) | 0.764 (0.080) | **23.406** (10.571) | 3 |
| DiffPIR | 0.536 | 0.484 | 0.496 | 3 | 28.41 (1.403) | 0.632 (0.061) | 26.376 (10.555) | 6 |
| DPS | 0.346 | 0.380 | 0.362 | 10 | 26.49 (1.550) | 0.540 (0.067) | 27.603 (11.127) | 11 |
| DAPS | 0.514 | 0.556 | 0.480 | 4 | 30.15 (1.429) | 0.725 (0.053) | 23.978 (10.630) | 4 |
| PnP-DM | 0.527 | **0.579** | **0.500** | **1** | 29.85 (2.934) | 0.730 (0.056) | 24.324 (10.413) | 5 |
| Fully sampled | 0.573 | 0.581 | 0.535 | – | – | – | 23.721 (10.824) | – |

For compressed sensing MRI, achieving good performance on general purpose metrics such as PSNR and SSIM is not always a sufficient signal for high-quality reconstruction, as hallucinations might lead to wrong diagnoses. We quantify the degree of hallucination by employing a pathology detector on the reconstructed images of different methods. Specifically, we finetune a medium-size YOLOv11 model (Khanam & Hussain, 2024) on a training set of fully sampled images with the fastMRI+ pathology annotations (Zhao et al., 2021) (22 classes in total). We calculate the mAP50 metric over the reconstructed results on 14 selected volumes with severe knee pathologies, which includes 171 test images in total. For each method, we report the Precision, Recall, and mAP50 metrics for detection, and PSNR, SSIM, and Data Misfit for reconstruction, as shown in Table 10. We also provide the rankings based on mAP50 and PSNR. Overall, the two rankings are correlated, which means that better pixel-wise accuracy indeed leads to a more accurate diagnosis. However, there are a few algorithms for which the two rankings disagree: Residual UNet, Score MRI, and RED-diff. The best methods for pathology detection are E2E-VarNet and PnP-DM.

## B INVERSE PROBLEM DETAILS

### B.1 LINEAR INVERSE SCATTERING

**Problem details** Consider a 2D object with permittivity distribution $\epsilon(\boldsymbol{r})$ in a bounded sample plane $\Omega \in \mathbb{R}^2$, which is immersed in the background medium with permittivity $\epsilon_b$. The permittivity contrast is given by $\Delta\epsilon(\boldsymbol{r}) = \epsilon(\boldsymbol{r}) - \epsilon_b$. At each time, the object is illuminated by an incident light field $\boldsymbol{u}_{\text{in}}(\boldsymbol{r})$ emitted by one of $N > 0$ transmitters, and the scattered light field $\boldsymbol{u}_{\text{sc}}(\boldsymbol{r})$ is measured by $M > 0$ receivers. We adopt the experimental setup in (Sun et al., 2018) where the transmitters and receivers are arranged along a circle $\Gamma \in \mathbb{R}^2$ that surrounds the object. Here, $\boldsymbol{r} := (x, y)$ denotes

the spatial coordinates. Under the first Born approximation (Wolf, 1969), the interaction between the light and object is governed by the following equation

$$\boldsymbol{u}_{\text{tot}}(\boldsymbol{r}) = \boldsymbol{u}_{\text{in}}(\boldsymbol{r}) + \int_{\Omega} g(\boldsymbol{r} - \boldsymbol{r}') \, f(\boldsymbol{r}') \, \boldsymbol{u}_{\text{in}}(\boldsymbol{r}') \, d\boldsymbol{r}', \quad \boldsymbol{r} \in \Omega, \tag{11}$$

where $\boldsymbol{u}_{\text{tot}}(\boldsymbol{r})$ is the total light field, and $f(\boldsymbol{r}) = \frac{1}{4\pi} k^2 \Delta\epsilon(\boldsymbol{r})$ is the scattering potential. Here, $k = 2\pi/\lambda$ is the wavenumber in free space, and $\lambda$ is the wavelength of the illumination. In the 2D space, the Green's function is given by

$$g(\boldsymbol{r} - \boldsymbol{r}') = \frac{i}{4} H_0^{(1)}(k_b \|\boldsymbol{r} - \boldsymbol{r}'\|_2) \tag{12}$$

where $k_b = \sqrt{\epsilon_b} k$ is the wavenumber of the background medium, and $H_0^{(1)}$ is the zero-order Hankel function of the first kind. Given the total field $\boldsymbol{u}_{\text{tot}}$ inside the sample domain $\Omega$, the scattered field at the sensor plane $\Gamma$ is given by

$$\boldsymbol{u}_{\text{sc}}(\boldsymbol{r}) = \int_{\Omega} g(\boldsymbol{r} - \boldsymbol{r}') \, f(\boldsymbol{r}') \, \boldsymbol{u}_{\text{tot}}(\boldsymbol{r}') \, d\boldsymbol{r}', \quad \boldsymbol{r} \in \Gamma. \tag{13}$$

Note that $\boldsymbol{r}$ denotes the sensor location in $\Gamma$, and the integral is computed over $\Omega$.

By discretizing Eq. (11) and Eq. (13), we obtain a vectorized system that describes the linear inverse scattering problem. We denote the 2D vectorized permittivity distribution of the object by $\boldsymbol{z} := f(\boldsymbol{r})$, and the corresponding measurement by $\boldsymbol{y}_{\text{sc}} = \boldsymbol{u}_{\text{sc}}(\boldsymbol{r})$ for notation consistency.

The forward model can thus be written as

$$\boldsymbol{y}_{\text{sc}} = \boldsymbol{H}(\boldsymbol{u}_{\text{tot}} \odot \boldsymbol{z}) + \boldsymbol{e} = \boldsymbol{A}\boldsymbol{z} + \boldsymbol{e}, \tag{14}$$

where $\boldsymbol{u}_{\text{tot}} = \boldsymbol{G}(\boldsymbol{u}_{\text{in}} \odot \boldsymbol{z})$, matrices $\boldsymbol{G}$ and $\boldsymbol{H}$ are discretizations of the Green's function at $\Gamma$ and $\Omega$, respectively, and $\boldsymbol{A} := \boldsymbol{H}\text{diag}(\boldsymbol{u}_{\text{tot}})$. We split and concatenate the real and imaginary part of $\boldsymbol{A}$, and pre-compute the singular value decomposition of $\boldsymbol{A}$ to facilitate the plug-and-play diffusion methods that exploit SVD of linear inverse problems.

We set the physical size of test images to 18cm×18cm, and the wavelength of the illumination to $\lambda = 0.84$cm as specified in (Sun et al., 2019). The forward model consists of $N = 20$ transmitters, placed uniformly on a circle of radius $R = 1.6$m. We further assume the background medium to be air with permittivity $\epsilon_b = 1$. We specify the number of receivers to be $M = 360, 180, 60$ in our experiments.

**Related work** Linear inverse scattering aims to reconstruct the spatial distribution of an object's dielectric permittivity from the measurements of the light it scatters (Wolf, 1969; Kak & Slaney, 2001). This problem arises in various applications, such as ultrasound imaging (Bronstein et al., 2002), optical microscopy (Choi et al., 2007; Sung et al., 2009), and digital holography (Brady et al., 2009). Due to the physical constraints on the number and placement of sensors, the problem is often ill-posed, as the scattered light field is undersampled. Linear inverse scattering is commonly formulated as a linear inverse problem using scattering models based on the first Born (Wolf, 1969) or Rytov (Devaney, 1981) approximations. These models enable efficient computation and facilitate the use of convex optimization algorithms. On the other hand, nonlinear approaches have been developed to image strongly scattering objects (Kamilov et al., 2015; Tian & Waller, 2015; Ma et al., 2018; Liu et al., 2018; Chen et al., 2020), although these methods generally have a higher computational complexity. Deep learning-based methods have also been explored for linear inverse scattering. A common approach is to train convolutional neural networks (CNNs) to directly invert the scattering process by learning an inverse mapping from the measurements to permittivity distribution (Sun et al., 2018; Li et al., 2018a;c; Wu et al., 2020). Recent research has extended these efforts to more advanced deep learning techniques, such as neural fields (Liu et al., 2022; Cao et al., 2024) and deep image priors (Zhou & Horstmeyer, 2020).

## B.2   COMPRESSED SENSING MULTI-COIL MRI

**Problem details** We use the raw multi-coil $k$-space data from the fastMRI knee dataset (Zbontar et al., 2018). We then estimate the coil sensitivity maps of each slice using the ESPIRiT (Uecker

et al., 2014) method implemented in `SigPy`[3]. Since different volumes in the dataset have different shapes, we adopt the preprocessing procedure in (Jalal et al., 2021), leading to images with $320 \times 320$ shape. The ground truth image is given by calculating the magnitude image of the Minimum Variance Unbiased Estimator (MVUE), which is used for all numbers reported in Table 4 and Table 5. The MVUE images are also used as ground truths for training the end-to-end deep learning methods `Residual UNet` and `E2E-VarNet` (Sriram et al., 2020).

**Related work**  Compressed sensing magnetic resonance imaging (CS-MRI) is a medical imaging technology that enables high-resolution visualization of human tissues with faster acquisition time than traditional MRI (Lustig et al., 2007). Instead of fully sampling the measurement space (a.k.a. $k$-space), CS-MRI only takes sparse measurements and then solves an inverse problem that recovers the underlying image (Lustig et al., 2008). The traditional approach is to solve a regularized optimization problem that involves a data-fit term and a regularization term, such as the total variation (TV) (Bouman & Sauer, 1993), and the $\ell_1$-norm after a sparsifying transformation, such as Wavelet transform (Ma et al., 2008) and dictionary decomposition (Ravishankar & Bresler, 2011; Huang et al., 2014; Zhan et al., 2015)). End-to-end deep learning methods have also demonstrated strong performance in MRI reconstruction. Prior works have proposed unrolled networks (Yang et al., 2016; Hammernik et al., 2017; Aggarwal et al., 2017; Schlemper et al., 2018; Liu et al., 2021), UNet-based networks (Lee et al., 2017; Hyun et al., 2017), GAN-based networks (Yang et al., 2018; Quan et al., 2018), among others (Wang et al., 2016; Zhu et al., 2018; Tezcan et al., 2018; Luo et al., 2019; Liu et al., 2020). These learning methods have achieved state-of-the-art performance on the fastMRI dataset (Zbontar et al., 2018). Another line of work is to employ image denoisers as plug-and-play prior (Liu et al., 2020; Jalal et al., 2021; Sun et al., 2024) Recently, diffusion model-based methods have been designed for CS-MRI reconstruction (Chung & Ye, 2022; Luo et al., 2022; Chung et al., 2023).

### B.3  BLACK HOLE IMAGING

**Problem details**  Measurements for black hole imaging are obtained using Very Long Baseline Interferometry (VLBI). The cross-correlation of the recorded scalar electric fields at two telescopes, referred to as the (ideal) *visibility*, is related to the ideal source image $\boldsymbol{z}$ through a Fourier transform, as given by the van Cittert-Zernike theorem:

$$\boldsymbol{I}_{(a,b)}^t(\boldsymbol{z}) = \int_\rho \int_\delta e^{-i2\pi\left(u_{(a,b)}^t\rho + v_{(a,b)}^t\delta\right)} \boldsymbol{z}(\rho,\delta)\mathrm{d}\rho\mathrm{d}\delta. \tag{15}$$

Here, $(\rho,\delta)$ denotes the angular coordinates of the source image, and $(u_{(a,b)}^t, v_{(a,b)}^t)$ is the dimensionless baseline vector between two telescopes $(a,b)$, orthogonal to the source direction. In practice, these measurements can be time-averaged over short intervals.

Due to atmospheric turbulence and instrumental calibration errors, the observed visibilities are corrupted by amplitude and phase errors, along with additive Gaussian thermal noise (EHTC, 2019a; Sun et al., 2024):

$$V_{(a,b)}^t = g_a^t g_b^t e^{-i(\phi_a^t - \phi_b^t)} \boldsymbol{I}_{(a,b)}^t(\boldsymbol{z}) + \boldsymbol{\eta}_{(a,b)}^t. \tag{16}$$

Note that there are three main sources of noise at time $t$: gain errors $g_a^t, g_b^t$, phase errors $\phi_a^t, \phi_b^t$, and baseline-based additive white Gaussian noise $\boldsymbol{\eta}_{(a,b)}^t$. While the phase of the visibility cannot be directly used due to phase errors, the product of three visibilities among any set of three telescopes, known as the *bispectrum*, can be computed to retain useful information. Specifically, the phase of the bispectrum, termed the *closure phase*, effectively cancels out phase errors in the observed visibilities. Similarly, a strategy can be employed to cancel out amplitude gain errors and extract information from the visibility amplitude (Blackburn et al., 2020). Formally, these quantities are defined as:

$$\boldsymbol{y}_{t,(a,b,c)}^{\mathrm{cp}} = \angle(V_{(a,b)}^t V_{(b,c)}^t V_{(a,c)}^t),$$

$$\boldsymbol{y}_{t,(a,b,c)}^{\mathrm{logca}} = \log\left(\frac{|V_{(a,b)}^t||V_{(c,d)}^t|}{|V_{(a,c)}^t||V_{(b,d)}^t|}\right). \tag{17}$$

---

[3]https://github.com/mikgroup/sigpy

Here, $\angle$ denotes the complex angle, and $|\cdot|$ computes the complex amplitude. For a total of $M$ telescopes, the number of closure phase measurements $\boldsymbol{y}^{\text{cp}}_{t,(a,b,c)}$ at time $t$ is $\frac{(M-1)(M-2)}{2}$, and the number of log closure amplitude measurements $\boldsymbol{y}^{\text{logca}}_{t,(a,b,c)}$ is $\frac{M(M-3)}{2}$, after accounting for redundancy. To avoid having to solve for the calibration terms, black hole imaging methods can constrain closure quantities during inference. Since closure quantities are nonlinear transformations of the visibilities, the forward model in black hole imaging then becomes non-convex.

The total flux of the image source, representing the DC component of the Fourier transform, is given by:

$$\boldsymbol{y}^{\text{flux}} = G^{\text{flux}}(\boldsymbol{z}) = \int_\rho \int_\delta \boldsymbol{z}(\rho, \delta) \mathrm{d}\rho \mathrm{d}\delta. \tag{18}$$

To aggregate data over time intervals and telescope combinations, the forward model of black hole imaging can be expressed as:

$$\boldsymbol{y} = G(\boldsymbol{z}, \xi) = \left( G^{\text{cp}}(\boldsymbol{z}), G^{\text{logca}}(\boldsymbol{z}), G^{\text{flux}}(\boldsymbol{z}) \right) = (\boldsymbol{y}^{\text{cp}}, \boldsymbol{y}^{\text{logca}}, \boldsymbol{y}^{\text{flux}}), \tag{19}$$

The data consistency is typically assessed using the $\boldsymbol{\chi}^2$ statistic:

$$\begin{aligned}
\mathcal{L}(\boldsymbol{y} \mid \boldsymbol{z}) &= \boldsymbol{\chi}^2_{cp} + \boldsymbol{\chi}^2_{\text{logca}} + \boldsymbol{\chi}^2_{\text{flux}} \\
&= \frac{1}{N_{\text{cp}}} \left\| \frac{G^{\text{cp}}(\boldsymbol{z}) - \boldsymbol{y}^{\text{cp}}}{\sigma_{\text{cp}}} \right\|^2 + \frac{1}{N_{\text{logca}}} \left\| \frac{G^{\text{logca}}(\boldsymbol{z}) - \boldsymbol{y}^{\text{logca}}}{\sigma_{\text{logca}}} \right\|^2 + \left\| \frac{G^{\text{flux}}(\boldsymbol{z}) - \boldsymbol{y}^{\text{flux}}}{\sigma_{\text{flux}}} \right\|^2.
\end{aligned} \tag{20}$$

Here, $\sigma_{\text{cp}}$, $\sigma_{\text{logca}}$, and $\sigma_{\text{flux}}$ are the estimated standard deviations of the measured closure phase, log closure amplitude, and flux, respectively. Additionally, $N_{\text{cp}}$ and $N_{\text{logca}}$ represent the total number of time intervals and telescope combinations for the closure phase and log closure amplitude measurements.

**Related work**  The Event Horizon Telescope (EHT) Collaboration aims to image black holes using a global network of radio telescopes operating at around a 1mm wavelength. Through very-long-baseline interferometry (VLBI) (Thompson et al., 2017), data from these telescopes are synchronized to obtain measurements in 2D Fourier space of the sky's image, known as *visibilities* (van Cittert, 1934; Zernike, 1938). These measurements only sparsely cover the low-spatial-frequency space and are corrupted by instrument noise and atmospheric turbulence, making the inverse problem of image recovery ill-posed. Using traditional imaging techniques, the EHT Collaboration has successfully imaged the supermassive black holes M87* (EHTC, 2019b; 2024) and SgrA* (EHTC, 2022). The classical imaging algorithm is CLEAN (Högbom, 1974; Clark, 1980), as implemented in the `DIFMAP` (Shepherd, 1997; 2011) software. `DIFMAP` is an inverse modeling approach that starts with the "dirty" image (given by the inverse Fourier transform of the visibilities) and iteratively deconvolves the image with an estimate point-spread function to "clean" the image. Since `DIFMAP` often requires a human-in-the-loop, we chose not to present results from `DIFMAP`. The EHT has also developed and used regularized maximum-likelihood approaches, namely `eht-imaging` (Chael et al., 2016; 2018; 2019) and `SMILI` (Akiyama et al., 2017b;a; 2019). Although they regularize and optimize the image differently (EHTC, 2019b), `eht-imaging` and `SMILI` both iteratively update an estimated image to agree with the measured data and regularization assumptions. Because of the simple regularization they choose to use, these baseline methods are limited in the amount of visual detail they can recover and do not recover detail far beyond the intrinsic resolution of the measurements. Some deep-learning-based regularization approaches have been proposed for VLBI (Feng et al., 2024; Feng & Bouman, 2024; Dia et al., 2023), but most plug-and-play inverse diffusion solvers have not been validated on black hole imaging.

**Multi-modal observation**  As previously discussed, the non-convex and sparse measurement characteristics of black hole imaging cause the inverse problem to exhibit multi-modal behavior. Consequently, the resulting samples may follow systematic modes that, while potentially quite different from the true source images, fit the observational data well and exhibit high prior likelihood. Figure 7 illustrates two such modes discovered by `DAPS` and `PnP-DM`. This multi-modal behavior has not been extensively formulated or discussed in previous literature, and we believe it represents a phenomenon worthy of further investigation.

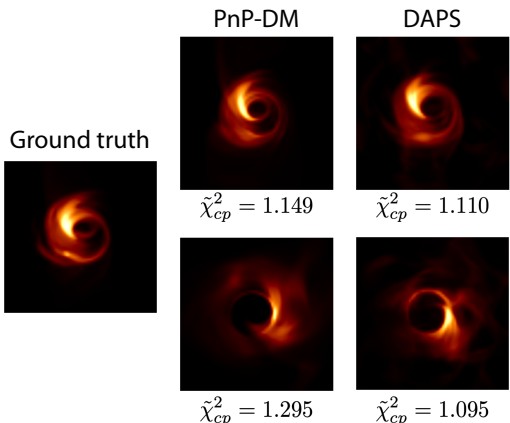

Figure 7: Multi-modal example on black hole imaging. The image shows two systematic modes discovered by `DAPS` and `PnP-DM`.

## B.4 FULL WAVEFORM INVERSION

**Problem details**   In Eq. (8), solving for the pressure wavefield $u$ given the velocity $v$ and source term $q$ defines the forward modeling process. We use an open-source software, `Devito` (Louboutin et al., 2019), for both forward modeling and adjoint FWI. We use a 128×128 mesh to discretize a physical domain of 2.54km × 1.27km, with a horizontal spacing of 20m and a vertical spacing of 10m. The time step is set to 0.001s which satisfies the Courant–Friedrichs–Lewy (CFL) condition (Courant et al., 1967). We use a Ricker wavelet with a central frequency of 5Hz to excite the wavefield and model it for 1s. The natural boundary condition is set for the top boundary (free surface) which will generate reflected waves, while the absorbing boundary condition (Clayton & Engquist, 1977) is set for the rest boundaries to avoid artificial reflections. The absorbing boundary width is set to 80 grid points.

Inferring the subsurface velocity from observed data at receivers defines the inverse problem. We put 129 receivers evenly near the free surface (at a depth of 10m) to model a realistic scenario. We excite 16 sources evenly at a depth of 1270m. This source-receiver geometry is designed for the entire physical medium to be sampled by seismic waves, making it theoretically feasible to invert for $v$. `Devito` uses the adjoint-state method to estimate the gradient by cross-correlating the forward and adjoint wavefields at zero time lag (Plessix, 2006):

$$\nabla_{\boldsymbol{m}}\Phi = \sum_{t=1}^{N_t} \boldsymbol{u}[t]\delta\boldsymbol{u}_{tt}[t] \quad \text{where} \quad \boldsymbol{m} = \frac{1}{\boldsymbol{z}^2}, \tag{21}$$

where $\Phi$ is the objective function, $N_t$ is the number of time steps, $\boldsymbol{u}$ is the forward wavefield, and $\delta\boldsymbol{u}$ is the adjoint wavefield. $\delta\boldsymbol{u}$ is generated by treating the receivers as sources and back-propagating the residual $\delta\boldsymbol{d}$ between the modeled and observed data into the model:

$$\begin{aligned} \boldsymbol{A}^T\delta\boldsymbol{u} &= \boldsymbol{P}^T\delta\boldsymbol{d}, \\ \delta\boldsymbol{d} &= \boldsymbol{y} - \boldsymbol{d}. \end{aligned} \tag{22}$$

This process is repeated for each source, and the gradients are summed to update the parameters of interest.

## B.5 NAVIER-STOKES

**Forward modeling**   The following 2D Navier-Stokes equation for a viscous, incompressible fluid in vorticity form on a torus defines the forward model.

$$\begin{aligned} \partial_t \boldsymbol{w}(\boldsymbol{x}, t) + \boldsymbol{u}(\boldsymbol{x}, t) \cdot \nabla \boldsymbol{w}(\boldsymbol{x}, t) &= \nu\Delta\boldsymbol{w}(\boldsymbol{x}, t) + f(\boldsymbol{x}), & \boldsymbol{x} &\in (0, 2\pi)^2, t \in (0, T] \\ \nabla \cdot \boldsymbol{u}(\boldsymbol{x}, t) &= 0, & \boldsymbol{x} &\in (0, 2\pi)^2, t \in [0, T] \\ \boldsymbol{w}(\boldsymbol{x}, 0) &= \boldsymbol{w}_0(\boldsymbol{x}), & \boldsymbol{x} &\in (0, 2\pi)^2 \end{aligned} \tag{23}$$

where $\boldsymbol{u} \in C\left([0,T]; H_{\mathrm{per}}^{r}((0,2\pi)^{2}; \mathbb{R}^{2})\right)$ for any $r > 0$ is the velocity field, $\boldsymbol{w} = \nabla \times \boldsymbol{u}$ is the vorticity, $\boldsymbol{w}_0 \in L_{\mathrm{per}}^2\left((0,2\pi)^2; \mathbb{R}\right)$ is the initial vorticity, $\nu \in \mathbb{R}_+$ is the viscosity coefficient, and $f \in L_{\mathrm{per}}^2\left((0,2\pi)^2; \mathbb{R}\right)$ is the forcing function. The solution operator $\mathcal{G} : \boldsymbol{w}_0 \rightarrow \boldsymbol{w}_T$ is defined as the operator mapping the vorticity from the initial vorticity to the vorticity at time $T$. We implement the forward model using a pseudo-spectral solver with adaptive time stepping (He & Sun, 2007).

**Dataset**  To generate the training and test samples, we first draw independent identically distributed samples from the Gaussian random field $\mathcal{N}\left(0, (-\Delta + 9\boldsymbol{I})^{-4}\right)$, where $-\Delta$ denotes the negative Laplacian. Then, we evolve them according to Equation 9 for 5 time units to get the final vorticity filed, which generates an empirical distribution of the vorticity field with rich flow features. We set the forcing function $f(\boldsymbol{x}) = -4\cos(4\boldsymbol{x}_2)$.

### B.6  PRETRAINED DIFFUSION MODEL DETAILS

We train diffusion models following the pipeline from (Karras et al., 2022), using UNet architectures from (Dhariwal & Nichol, 2021) and (Song et al., 2020). Detailed network configurations can be found in Table 11.

Table 11: Model Card for pre-trained diffusion models.

|  | Inverse scattering | Black hole | MRI | FWI | 2D Navier-Stokes |
|---|---|---|---|---|---|
| Input resolution | $128 \times 128$ | $64 \times 64$ | $2 \times 320 \times 320$ | $128 \times 128$ | $128 \times 128$ |
| # Attention blocks in encoder/decoder | 5 | 3 | 5 | 5 | 5 |
| # Residual blocks per resolution | 1 | 1 | 1 | 1 | 1 |
| Attention resolutions | $\{16\}$ | $\{16\}$ | $\{16\}$ | $\{16\}$ | $\{16\}$ |
| # Parameters | 26.8M | 20.0M | 26.0M | 26.8M | 26.8M |
| # Training steps | 50,000 | 50,000 | 100,000 | 50,000 | 50,000 |

### B.7  ALGORITHMS AND PARAMETER CHOICES

#### B.7.1  PROBLEM-SPECIFIC BASELINES

**Black hole imaging**  We use `SMILI` (Akiyama et al., 2017b;a; 2019) and `eht-imaging` (Chael et al., 2016; 2018; 2019) as our baseline methods. To ensure compatibility with the default hyperparameters of these methods, we preprocess the test dataset accordingly.

**Full waveform inversion**  A classic baseline for full waveform inversion is `LBFGS`. We set the maximum iteration to 5 and perform 100 global update steps with a Wolfe line search. The second baseline we consider is the `Adam` optimization algorithm (Kingma, 2014). We implement the `Adam` optimizer with a learning rate of 0.02 with the learning rate decay to minimize the data misfit term. For the traditional method, the initialization is a smoothed version of the ground truth, which is blurred using a Gaussian filter with $\sigma = 20$. We perform the inversion for 300 iterations.

**Linear inverse scattering**  We include `FISTA-TV` (Sun et al., 2019) as a traditional optimization-based method. We set batch size $B = 20$ and $\tau = 5 \times 10^{-7}$ for all experiments.

**Compressed sensing multi-coil MRI**  We utilize both traditional methods, such as `Wavelet+`$\ell_1$ (Lustig et al., 2007; Lustig et al., 2008) and `TV`, as well as end-to-end models like `Residual UNet` and `E2E-VarNet` (Sriram et al., 2020). For the traditional methods, we apply the same hyperparameter search strategy for fine-tuning, while the end-to-end models are trained using the `Adam` optimizer with a learning rate of $1 \times 10^{-4}$ until convergence.

**Navier-Stokes equation**  The traditional baseline we implement is the Ensemble Kalman Inversion (EKI) first proposed in Iglesias et al. (2013). It is implemented with 2048 particles, 500 update steps, and adaptive step size used in (Kovachki & Stuart, 2019) to ensure similar computation budget. Additional baselines include `DPS-fGSG` and `DPS-cGSG`, which are natural `DPS` extensions that replace gradient by zeroth-order gradient estimation first introduced in Zheng et al. (2024). More

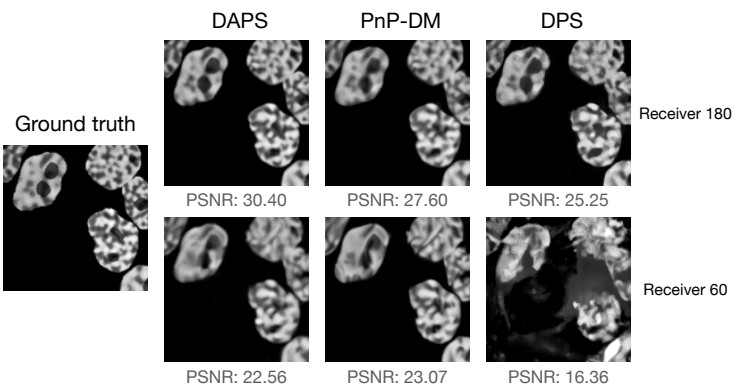

Figure 8: Robustness to a human face prior in linear inverse scattering shows that methods requiring more data gradient steps tend to generalize better than those that prioritize the prior more.

specifically, we use forward and central Gaussian Smoothed Gradient estimation technique (Berahas et al., 2022).

### B.7.2 HYPERPARAMETER SELECTION

To ensure sufficient tuning of the hyperparameters for each algorithm, we employ a hybrid strategy combining grid search with Bayesian optimization and early termination technique, using a small validation dataset. Specifically, we first perform a coarse grid search to narrow down the search space and then apply Bayesian optimization. For problems where the forward model is fast such as linear inverse scattering, MRI, and black hole imaging, we conduct 50-100 iterations of Bayesian optimization to select the best hyperparameters. For computationally intensive problems such as full waveform inversion and Navier-Stokes equation, we use 10-30 iterations of Bayesian optimization combined with an early termination technique (Li et al., 2018b), based on data misfit. The details of the search spaces for Bayesian optimization and the optimized hyperparameter choices are listed in Table 12.

## C  FUTURE DIRECTION

In this section, we outline a few additional future directions for benchmarking PnPDP methods for solving inverse problems.

**Robustness to forward model mismatch**   In our paper, we only consider the setting where the forward model is exact and explicit. However, in real-world scenarios, the forward model is often an approximation. Studying the robustness of algorithms when there is a mismatch between the assumed and actual forward model would be an essential step toward their practical deployment. This includes exploring how algorithms handle noisy or imperfect forward model representations.

**Robustness to prior mismatch**   The robustness of a diffusion prior often depends on two key factors: the degree of ill-posedness and the balance between the prior and the observation in the sampling algorithm. Highly ill-posed tasks, such as black hole imaging, require a more in-distribution prior to achieve reasonable results, whereas less ill-posed tasks, such as linear inverse scattering, are less sensitive to this requirement. Regarding the balance between prior and observation, methods like DAPS and PnP-DM, which incorporate more data gradient steps, tend to be more robust to out-of-distribution prior than methods like DPS. We include a preliminary discussion on this in Figure 8 presents a comparison of the robustness of different algorithms in the linear inverse scattering using a prior trained on FFHQ 256×256 human face images. We will pursue further exploration in this direction in the future.

**Multi-modal posterior**   In this work, we focus exclusively on scenarios where the ground truth is unimodal. However, some real-world inverse problems might exhibit posteriors that are inherently

Table 12: Hyperparameter search space and final choices of the diffusion-model-based algorithms on all five inverse problems. Columns marked with task names present the chosen values for the reported main results in Appendix A.1. These values are selected by a hybrid hyperparameter search strategy described in Appendix B.7.2.

| Methods/Parameters | Search space | Linear inverse scattering (360 / 180 / 60) | Black hole | MRI (Sim. / Raw) | FWI | 2D Navier-Stokes |
|---|---|---|---|---|---|---|
| **DPS** | | | | | | |
| Guidance scale | $[10^{-3}, 10^3]$ | 280/380/625 | 0.003 | 0.589/0.428 | $10^{-2}$ | – |
| **LGD** | | | | | | |
| Guidance scale | $[10^{-3}, 10^4]$ | 3200/6400/13000 | 0.0082 | – | 11.73 | – |
| # MC samples | $[1, 20]$ | 20 | 8 | – | 5 | – |
| **REDDiff** | | | | | | |
| Learning rate | $[10^{-4}, 1.0]$ | 0.04 | 0.05 | $4 \times 10^{-2}$ / $2.96 \times 10^{-2}$ | 0.01 | – |
| Regularization $\lambda_{\text{base}}$ | $[10^{-3}, 1.0]$ | 0.0005 | 0.25 | $2.33 \times 10^{-1}$ / $2.72 \times 10^{-3}$ | 0.1 | – |
| Regularization schedule | constant, linear, sqrt | constant | constant | sqrt | linear | – |
| Gradient weight | $[10^{-2}, 10^2]$ | 1500 | 0.0004 | $6.68 \times 10^1$ / $1.7 \times 10^{-2}$ | 1 | – |
| **DiffPIR** | | | | | | |
| # sampling steps | $\{200, 400, \ldots, 1000\}$ | 200 | 1000 | 1000 | 1000 | – |
| Regularization $\lambda$ | $[1, 10^5]$ | $4 \times 10^{-4}$/$2 \times 10^{-4}$/$10^{-4}$ | 113.6 | 163 / 1.31 | 80.6 | – |
| Stochasticity $\zeta$ | $[10^{-5}, 1]$ | 1 | 0.34 | 0.114 / 0.478 | 0.11 | – |
| Noise level $\sigma_y$ | $[10^{-2}, 10^1]$ | 0.01 | 1.4 | $1.05 \times 10^{-2}$ / $1.36 \times 10^{-1}$ | 0.28 | – |
| **PnPDM** | | | | | | |
| Annealing step | $[50, 200]$ | 100 | 100 | 100 | 150 | – |
| Annealing sigma max | $[10, 50]$ | 10 | 10 | 10 | 25 | – |
| Annealing decay rate | $[0.60, 0.99]$ | 0.9 | 0.93 | 0.93 | 0.99 | – |
| Langevin step size | $[10^{-6}, 10^{-3}]$ | $2 \times 10^{-5}$/$4 \times 10^{-5}$/$10^{-4}$ | $10^{-5}$ | $10^{-6}$ | $3 \times 10^{-4}$ | – |
| Langevin step number | $[10, 500]$ | 200 | 200 | 200 | 10 | – |
| Noise level | $[10^{-4}, 10^1]$ | $10^{-4}$ | 1 | $1.02 \times 10^{-3}$ / $1.15 \times 10^{-2}$ | 1 | – |
| **DAPS** | | | | | | |
| Annealing step | $[50, 200]$ | 200 | 100 | 200 | 150 | – |
| Diffusion step | $[1, 10]$ | 10 | 5 | 5 | 5 | – |
| Langevin step size | $[10^{-6}, 10^{-3}]$ | $4 \times 10^{-5}$/$8 \times 10^{-5}$/$2 \times 10^{-4}$ | $10^{-4}$ | $1.03 \times 10^{-5}$ / $1.52 \times 10^{-5}$ | $3 \times 10^{-4}$ | - |
| Langevin step number | $[10, 500]$ | 50 | 20 | 100 | 50 | - |
| Noise level | $[10^{-4}, 10^1]$ | $10^{-4}$ | 1 | $1.63 \times 10^{-3}$ / $4.77 \times 10^{-3}$ | 1 | – |
| Step size decay | $[0.1, 1]$ | 1/1/0.5 | 1 | 1 | 1 | - |
| **DDRM** | | | | | | |
| Stochasticity $\eta$ | $[0, 1]$ | 0.85 | – | – | – | – |
| **DDNM** | | | | | | |
| Stochasticity $\eta$ | $[0, 1]$ | 0.95 | – | – | – | – |
| # time-travel steps $L$ | $[0, 5]$ | 1 | – | – | – | – |
| **ΠGDM** | | | | | | |
| Stochasticity $\eta$ | $[0, 1]$ | 0.2 | – | – | – | – |
| **FPS** | | | | | | |
| Stochasticity $\eta$ | $[0, 1]$ | 0.9 | – | – | – | – |
| # particles | $[1, 20]$ | 20 | – | – | – | – |
| **MCG-diff** | | | | | | |
| # particles | $[1, 64]$ | 16 | – | – | – | – |
| **DPS-fGSG** | | | | | | |
| Guidance scale | $[10^{-2}, 10^2]$ | – | – | – | – | 0.1 |
| **DPS-cGSG** | | | | | | |
| Guidance scale | $[10^{-2}, 10^2]$ | – | – | – | – | 0.1 |
| **DPG** | | | | | | |
| # MC samples | $\{1000, 2000, \ldots, 6000\}$ | – | – | – | – | 4000 |
| Guidance scale | $[10^{-1}, 10^3]$ | – | – | – | – | 64 |
| **SCG** | | | | | | |
| # MC samples | $\{128, 256, 512\}$ | – | – | – | – | 512 |
| **EnKG** | | | | | | |
| Guidance scale | $\{1.0, 2.0, 4.0\}$ | - | – | – | – | 2.0 |
| # particles | $\{512, 1024, 2048\}$ | - | – | – | – | 2048 |

multimodal, reflecting multiple plausible solutions given the observations. Developing systematic benchmarks to evaluate how accurately different PnPDP methods can capture such multimodal posteriors is an interesting and challenging research question.

