# OpenReview forum: "InverseBench: Benchmarking Plug-and-Play Diffusion Priors for Inverse Problems in Physical Sciences"
_ICLR.cc/2025/Conference — ICLR 2025 Spotlight_

### Official Review · Reviewer_MUia · 2024-10-24

**Soundness:** 4
**Presentation:** 4
**Contribution:** 3
**Rating:** 8
**Confidence:** 4

**Summary:**

The paper provides a comprehensive summary of the performance of 14 different diffusion-based inverse problem solvers on 5 scientific applications, each with unique forward operators. Extensive experiments on each application reveal general trends such as a bias towards the prior with out-of-distribution data and strong performance compared with conventional baselines. They contribute an open-source codebase of datasets and pretrained models to enable expedited exploration in this space of inverse problems.

**Strengths:**

- With such a wide and rapidly expanding collection of diffusion-based methods, this paper provides a valuable assessment of existing models across diverse problems. For those working in the space of inverse imaging problems, this gives a strong starting point to assess which model to use for a given application or as a baseline.
- The experiments are extensive and thorough. As such, they are able to discover high-level insights and trends across problems. This breadth of experimentation would be larger unavailable to most researchers; thus, the results present a significant contribution to the community in my view.
- Open-sourced datasets and pretrained models contribute a well-curated foundation for other researchers to build upon
- The paper is very well-written and concise with clear introductions to the complex inverse problems and well-thought-out analyses of the results.

**Weaknesses:**

- The value of some of the images in Figure 2 is limited since its difficult to actually see differences between the reconstructions. Particularly, the MRI reconstructions for the PnP methods look the exact same. It may be more useful to provide a zoomed-in region where the differences are more apparent.

Minor Points:
- L161: Typo with "... generally intractable as so various approximations..."
- L508: Typo with "inverse scattering on sources that contain more than cells..."

**Questions:**

- Are the reported results (say in Table 4) for a single posterior sample? Or do you draw many samples and compute a posterior mean?

---

> ### Author Response · Authors · 2024-11-21
>
> We thank the reviewer for the positive feedback and constructive suggestions. We address the question below and incorporate the reviewer's suggestions below.
>
> **W1**:
> We have updated Figure 2 with zoom-in regions and error maps for MRI reconstructions to highlight the differences between them.
>
> **W2**:
> The typos are corrected in the revised version.
>
> **Q1**:
> For most algorithms, the reported results, including those in Table 4, are based on a single posterior sample for each test instance. Specifically, we obtain one sample from the algorithm per test instance and compute the mean and standard deviation of the evaluation metrics overall test samples.
> The only exceptions are ensemble-based methods such as EnKG and EKI, which operate with an ensemble of particles. For these methods, we compute the average error across the ensemble members for each test instance and then report the aggregate statistics over the test set.

---

> > ### Comment · Reviewer_MUia · 2024-11-21
> >
> > The authors have addressed all of my concerns and questions. As a result, I will keep my rating of an 8 as it is.

---

### Official Review · Reviewer_SNSr · 2024-10-30

**Soundness:** 4
**Presentation:** 4
**Contribution:** 3
**Rating:** 8
**Confidence:** 4

**Summary:**

Authors propose InverseBench, a benchmark/framework for evaluating diffusion prior based inverse problem solvers across five distinct scientific inverse problems that arise from black hole imaging, optical tomography, etc. Each of these problems present unique challenges for reconstruction that is different than the existing benchmarks. Authors benchmark 14 diffusion based algorithms that can be grouped into 4 categories (Guidance based, Variable splitting, Variational Bayes, Sequential Monte Carlo) and compare against problem specific baselines.

**Strengths:**

* The paper is written very well. I truly enjoyed reading it.
* Having a standardized benchmark to evaluate diffusion prior based algorithms is an important contribution to the inverse problems community. I believe the forward models that are included in the InverseBench are diverse, practically relevant with diverse characteristics (linear vs non-linear, with or without SVD decomposition, etc.).
* The diffusion algorithms that are considered in this paper are comprehensive.

**Weaknesses:**

* While I appreciate the comprehensiveness of experiments. I would expect more detailed discussion on experimental results. For instance, which methods (or category of methods) work better for each task based on the experimental results? If a method is competitive when used in natural image inpainting, super-resolution but not in InverseBench, is there some intuition/explanation why that might be the case?
* See the questions below.

**Questions:**

* Line 316: "Since this dataset is not publicly available, we generate synthetic images from a pre-trained diffusion modes..." Do you use the model that is trained using 50k images to synthetically generate 105 more images for validation and test purposes? In general this part is not clear to me.
* As a future direction, robustness of diffusion prior methods can be systematically investigated. That is suppose measurement is obtained via forward model $G$ but reconstructed believing that it came from
$\tilde{G}$. The experiments conducted on CS-MRI already suggests that PnP diffusion prior methods are more robust than baseline and end-to-end methods but it is also interesting to see comparison between different PnPDP algorithm.
* Have the authors considered benchmarking diffusion model checkpoints that are pre-trained on natural images (such as ImageNet) on these tasks? It would be interesting to see if natural image priors are useful for the tasks in the InverseBench.

---

> ### Author Response · Authors · 2024-11-21
>
> **W1**:
> We thank the reviewer for the feedback. In response, we highlight two examples from our benchmarks, DAPS and PnP-DM, that illustrate the nuances of performance differences between natural image restoration tasks and the scientific inverse problems.
>
> Both DAPS and PnP-DM achieve near state-of-the-art results on natural image restoration tasks such as inpainting and super-resolution. However, their performance significantly deteriorates in problems like Full Waveform Inversion (FWI) in InverseBench. This issue stems from their inability to account for the stability conditions required to query the forward model. For example, the numerical PDE solvers used in FWI and the Navier-Stokes equations impose strict stability constraints, such as the Courant–Friedrichs–Lewy (CFL) condition. The Langevin Monte Carlo (LMC) subroutine used by DAPS and PnP-DM introduces noisy perturbations in the input. While such noise is tolerable in natural image restoration tasks, it violates the stability conditions in these scientific inverse problems, resulting in numerical instability and unreliable gradient estimates. We discuss this in detail in lines 435–451 of the paper, where we also provide examples of these instabilities.
>
>
> **Q1**:
> Thank you for pointing this out; your understanding is correct. We employ a separate model, initialized with different random seeds, trained on a dataset of 50,000 non-publicly available images. This model is used to unconditionally generate 5 images for the validation dataset and 100 images for the test dataset. These generated datasets are utilized for hyperparameter tuning and evaluation.
>
> **Q2**:
> We certainly agree that investigating robustness is very important.  As alluded to by your question, there are two main types of robustness that are important to study: prior mismatch and forward model G mismatch.  We have added a discussion with some of our preliminary results on this research direction in Appendix B.
>
> **Q3**:
> This is a good question. We have experimented with FFHQ and ImageNet pretrained priors on black hole imaging and inverse scattering tasks. Our findings are summarized as follows: the more ill-posed a problem is, the more critical the role of the prior becomes.
>
> Because black hole imaging is highly ill-posed, combining the FFHQ pretrained prior with PnP diffusion methods produces an image resembling a human face while still satisfying the observational black hole data with a $\chi^2$ value between 1.1 and 1.5. However, these images significantly diverge from the true black hole images, with a substantial PSNR drop of about 5–8 dB compared to in-distribution data priors. However, for the less ill-posed inverse scattering task, comparable results can be achieved between the baseline and the PnP diffusion with in-distribution priors. The table below presents the evaluation results for inverse scattering on 100 test samples. We can observe that both the FFHQ and ImageNet priors improve the results compared to the baseline, which uses only total variation (TV) as the prior.
>
>
> | Methods/inverse scattering receivers | 360    | 180    | 60     |
> | ------------------------------------ | ------ | ------ | ------ |
> | DAPS + In-distribution Data Prior    | 34.641 | 33.160 | 25.875 |
> | DAPS + FFHQ Prior                    | 33.048 | 30.821 | 21.413 |
> | DAPS + ImageNet Prior                | 32.873 | 30.412 | 21.223 |
> | FISTA+TV                             | 32.126 | 26.523 | 20.938 |
>
> An important implementation detail to highlight is the mismatch between the dimensions of the priors and the dimensions of the tasks. For example, the typical dimension of an ImageNet pretrained prior is (3, 256, 256), while the dimension for inverse scattering is (1, 128, 128). To resolve this mismatch, we introduced a wrapper before the forward function of each task to map the prior dimensions to the task dimensions. Specifically, we downsampled the images and converted them to grayscale, aligning the (3, 256, 256) prior dimensions with the (1, 128, 128) task dimensions used in inverse scattering.

---

> > ### Comment · Reviewer_SNSr · 2024-11-25
> >
> > I would like to thank the authors for their detailed responses and providing the table with FFHQ/ImageNet priors. My questions and concerns are addressed. Therefore, I am happy to increase my score to $8$.

---

### Official Review · Reviewer_TH8o · 2024-11-04

**Soundness:** 3
**Presentation:** 3
**Contribution:** 3
**Rating:** 6
**Confidence:** 4

**Summary:**

This submission made technical contributions by creating and demonstrating the INVERSEBEHCN framework, which is designed for evaluating plug-and-play diffusion models on scientific inverse problems such as black hole imaging, fluid dynamics, medical imaging, and so on. The authors compared 14 representative methods in different problem settings (i.e., forward model), considering the computation efficiency and solution accuracy. What's more, the adaptability of each method to different types of forward models was investigated.

**Strengths:**

The proposed INVERSEBECH fills a gap in testing diffusion models beyond natural image tasks by evaluating them to scientifically meaningful, diverse, and challenging problems. The selected 14 methods are evaluated in multiple perspectives, such as efficiency, accuracy and adaptability.

**Weaknesses:**

1. The discussion on sensitivity to hyperparameters and initialization for each method is limited.

2. The investigation into out-of-distribution could be more elaborated, as this could usually happen in the real-world inverse problem. Would a better sampling method help us to overcome the problem caused by out-of-distribution, or is this more related to trained prior?

3. Referring to the selected 14 methods as "plug-and-play" methods is potentially misleading, especially since one of them is specifically a plug-and-play diffusion method (Wu et al., 2024). This raises the question of whether "plug-and-play" should be considered a more narrowly defined concept.

**Questions:**

1. How to make sure that each method is optimally tuned for different types of inverse problems, such as implementation details, hyperparameter choice? Even we understand that this would be a huge amount of work to tune every method.

2. As shown in Table 11, methods like Reddiff, DiffPIR, PnPDM, and DAPS have more hyperparameters for tuning than other methods. Meanwhile, these methods tend to have better positions in the benchmark shown in Figure 1. Could this suggest that the other methods are genuinely worse or that they may be undertuned?

3. For the domain-specific evaluation like compressed sensing MRI, the papers that apply generative models to MRI reconstruction in a plug-and-play way have been published in some domain-specific journals. To reach a broader impact, it is reasonable to briefly introduce the following works.

     1. Tezcan, Kerem C., et al. "MR image reconstruction using deep density priors." IEEE transactions on medical imaging 38.7 (2018): 1633-1642.
     2. Luo, Guanxiong, et al. "MRI reconstruction using deep Bayesian estimation." Magnetic resonance in medicine 84.4 (2020): 2246-2261.
     3. Luo, Guanxiong, et al. "Bayesian MRI reconstruction with joint uncertainty estimation using diffusion models." Magnetic Resonance in Medicine 90.1 (2023): 295-311.

---

> ### Author Response · Authors · 2024-11-21
>
> We appreciate the constructive feedback and references. We address the reviewer's questions and concerns below.
>
> **W1 & Q1**:
> To ensure sufficient tuning of the hyperparameters for each algorithm, we employ a two-stage strategy combining grid search with Bayesian optimization and an early termination technique, using a small validation dataset. Specifically, we first perform a coarse grid search to narrow down the search space and then apply Bayesian optimization. For problems where the forward model is fast, such as linear inverse scattering, MRI, and black hole imaging, we conduct 50-100 iterations of Bayesian optimization to select the best hyperparameters. For computationally intensive problems such as full waveform inversion and Navier-Stokes equation, we use 10-30 iterations of Bayesian optimization combined with an early termination technique (Hyperband stopping algorithm [1]). We have added a discussion on hyperparameters and initialization to Appendix A.4.2 in the revised version.
>
> **W2**:
> In general, we observe three key aspects that can influence out-of-distribution performance: (1) the ill-posedness of the task, (2) the relevance of the trained prior, and (3) the balance between the prior and observation in the sampling algorithm. Tasks that are more ill-posed, use less relevant priors, or involve fewer data gradient steps typically exhibit worse generalization performance. Additional discussions on this topic are provided in Appendix B’s robustness section.
>
> **W3**:
> We agree that the naming can be confusing since one of the methods has “Plug-and-Play” in its title.  Nonetheless, this general class of methods is commonly known as plug-and-play methods [2].To avoid confusion, we use different fonts when referring to a specific algorithm like `PnP-DM` in the main text throughout the paper (e.g. line 442). We have added a clarification on the difference between PnPDP and `PnP-DM` in lines 155-156 to avoid confusion.
>
>
> **Q2**:
> We appreciate the reviewer’s observation regarding the relationship between hyperparameter space and the relative ranking of algorithms. As noted in our response to W1, we employ a tuning strategy to ensure fair comparisons and sufficient optimization for all methods.
>
> The stronger performance of RED-diff, DiffPIR, DAPS, and PnP-DM can likely be attributed to their algorithmic design, which leverages more sophisticated mechanisms that inherently require more parameters to tune. However, a larger hyperparameter space does not always guarantee better performance. For example, RED-diff, PnP-DM, and DAPS perform poorly on full waveform inversion as shown in Figure 1. This suggests that the relationship between the size of the hyperparameter space and performance is task-dependent and not straightforward. While larger parameter spaces can offer greater flexibility, performance ultimately depends on how well the algorithm aligns with the structural requirements of each specific inverse problem.
>
> **Q3**:
> Thanks for bringing these works to our attention. We have included these papers in the discussion on related works for MRI in Appendix A.2.2.
>
> [1] Li, Lisha, et al. "Hyperband: A novel bandit-based approach to hyperparameter optimization." Journal of Machine Learning Research 18.185 (2018): 1-52.
>
> [2]: Venkatakrishnan, Singanallur V., Charles A. Bouman, and Brendt Wohlberg. "Plug-and-play priors for model based reconstruction." 2013 IEEE global conference on signal and information processing. IEEE, 2013.

---

> ### Comment · Reviewer_TH8o · 2024-11-25
>
> I would like to thank the authors for their efforts in the rebuttal. Your answers and further explanations have effectively addressed my initial questions. However, one additional point slipped my mind when I was drafting my earlier reviews: How do you ensure that the priors used for comparison across all methods have the same capability? I am sorry for the delay in raising this question.

---

> > ### Author Response · Authors · 2024-11-26
> >
> > We thank the reviewer for the kind words and are glad that our previous responses have addressed the initial concerns. Regarding the additional question about the prior, we use the same pre-trained diffusion model across all different PnP diffusion prior methods for fair comparison. The configurations of the pre-trained diffusion models for each problem are detailed in Table 10. These models were trained using the EDM [1] framework on the training sets defined in Section 4.1.
> >
> > By standardizing the diffusion priors, we aim to isolate and evaluate the performance differences arising from the methods rather than from variations in the priors. We hope this addresses the reviewer’s question and clarifies our approach.
> >
> > [1]: Karras, Tero, et al. "Elucidating the design space of diffusion-based generative models." Advances in neural information processing systems 35 (2022): 26565-26577.

---

> > ### Author Response · Authors · 2024-12-02
> > **Follow-Up on the additional question**
> >
> > Dear Reviewer TH8o,
> >
> > Thank you once again for your thoughtful review and valuable feedback on our manuscript.
> >
> > As the discussion period concludes today, we wanted to kindly follow up to ensure that our responses have addressed your follow-up question. If there are any remaining concerns or clarifications needed, we are happy to address them promptly.
> >
> > We also want to kindly ask if it would be possible to update the score to reflect the entire discussion period. We sincerely value your input and the insights you’ve provided throughout this process.
> >
> > Best regards

---

### Official Review · Reviewer_69mk · 2024-11-04

**Soundness:** 3
**Presentation:** 4
**Contribution:** 3
**Rating:** 8
**Confidence:** 4

**Summary:**

A paper entitled “INVERSEBENCH: BENCHMARKING PLUG-AND-PLAY DIFFUSION MODELS FOR SCIENTIFIC INVERSE PROBLEMS” proposes a systematic benchmark for a number of scientific inverse problems that go beyond natural mages. The authors suggest optical tomography, black hole imaging, medical imaging, seismology, and fluid dynamics as a set of tasks for the benchmark. Next, the authors evaluate a handful of SOTA models with respect to their efficiency and accuracy on respective inverse tasks.

**Strengths:**

- This is an important and timely contribution. It has the potential to improve generalisation and standardisation across solutions addressing inverse problems in science.

**Weaknesses:**

- Using SSIM and PSNR is not always sufficient in medical imaging. Perhaps the authors could propose an approach to quantify the contribution of hallucinations.
- Although the title claims “Scientific Inverse Problems”, the collection of problems is extremely physics-heavy. The only non-physics problem the authors are looking at is MRI, I suggest toning down the claim in the title.

**Questions:**

Perhaps the authors could propose an approach to quantify the contribution of hallucinations? This is especially important for medical images, as hallucinations can lead to a wrong diagnosis.

---

> ### Author Response · Authors · 2024-11-21
>
> **W1 & Q1**:
> Thank you for your constructive feedback. We agree that metrics like PSNR and SSIM do not fully reflect the quality of MRI reconstructions, especially for diagnostic purposes. Therefore, we propose to quantify the degree of hallucination by employing a pathology detector and calculating the mAP50 metric over reconstructions. The results are summarized in the table below. For each method, we report the Precision, Recall, and mAP50 metrics for detection, and PSNR, SSIM, and Data Misfit for reconstruction. We also provide the rankings based on mAP50 and PSNR. Overall, the two rankings are correlated, which means that better pixel-wise accuracy indeed leads to a more accurate diagnosis. However, there are a few algorithms for which the two rankings disagree: Residual UNet, Score MRI, and RED-diff. The best methods are rE2E-VarNet and PnP-DM.
>
> Details: 1) The detector is finetuned based on a medium-sized YOLOv11 model on a training set of fully sampled images with the fastMRI+ pathology annotations (22 classes in total) [2]. 2)The detection metrics are calculated based on 14 selected volumes with severe knee pathologies, which lead to 171 test images in total.
>
> | Method        | Wavelet+$\ell_1$ | TV              | Residual UNet | E2E-VarNet        | CSGM            | ScoreMRI        | RED-diff            | DiffPIR         | DPS             | DAPS            | PnP-DM          | Fully sampled   |
> |---------------|------------------|-----------------|---------------|-------------------|-----------------|-----------------|---------------------|-----------------|-----------------|-----------------|-----------------|-----------------|
> | Precision     |            0.532 |           0.447 |         0.482 |         **0.610** |           0.501 |           0.412 |               0.478 |           0.536 |           0.346 |           0.514 |           0.527 |           0.573 |
> | Recall        |            0.332 |           0.251 |         0.462 |             0.514 |           0.528 |           0.554 |               0.468 |           0.484 |           0.380 |           0.556 |       **0.579** |           0.581 |
> | mAP50         |            0.385 |           0.263 |         0.439 |         **0.500** |           0.454 |           0.470 |               0.448 |           0.496 |           0.362 |           0.480 |       **0.500** |           0.535 |
> | mAP50 ranking |                9 |              11 |             8 |             **1** |               6 |               5 |                   7 |               3 |              10 |               4 |           **1** |              -- |
> | PSNR          |    28.16 (1.724) |   28.31 (1.834) | 31.62 (1.635) | **32.25** (1.901) |   27.34 (2.770) |   26.86 (2.583) |       31.56 (2.337) |   28.41 (1.403) |   26.49 (1.550) |   30.15 (1.429) |   29.85 (2.934) |              -- |
> | SSIM          |    0.685 (0.064) |   0.662 (0.079) | 0.803 (0.050) | **0.805** (0.056) |   0.673 (0.082) |   0.547 (0.092) |       0.764 (0.080) |   0.632 (0.061) |   0.540 (0.067) |   0.725 (0.053) |   0.730 (0.056) |              -- |
> | Data misfit   |  23.501 (10.475) | 24.182 (10.613) |            -- |                -- | 23.483 (10.651) | 25.677 (10.491) | **23.406** (10.571) | 26.376 (10.555) | 27.603 (11.127) | 23.978 (10.630) | 24.324 (10.413) | 23.721 (10.824) |
> | PSNR ranking  |                8 |               7 |             2 |             **1** |               9 |              10 |                   3 |               6 |              11 |               4 |               5 |              -- |
>
> **W2**:
> We appreciate the reviewer’s feedback on the scope of our work. While we acknowledge that InverseBench focuses on a subset of scientific inverse problems, we note that the selected problems span very diverse scientific domains, including black hole imaging, seismology, optical tomography, medical imaging (MRI), and fluid dynamics. Importantly, these tasks were chosen to represent a broad spectrum of problem characteristics encountered in scientific inverse problems, as outlined in Table 2.
>
> Regarding the reviewer’s question about “physics-heavy”, we note that most scientific inverse problems inherently involve physical modeling, as they often arise from real-world systems governed by physical laws. Even in the case of MRI, the forward model is based on the Bloch equations, which are also from physics principles.
>
> If the reviewer’s concern relates to the emphasis on problems with explicit forward models for measurements, we could clarify this scope in the title by replacing “Scientific Inverse Problems” with “Model-based Scientific Inverse Problems.” We welcome the reviewer’s feedback on this proposed revision.

---

> > ### Author Response · Authors · 2024-11-21
> > **References**
> >
> > [1]: Hyungjin Chung and Jong Chul Ye. Score-based diffusion models for accelerated MRI. Medical Image Analysis, 80:102479, 05 (2022).
> >
> > [2]: Ruiyang Zhao, et al. fastMRI+, Clinical pathology annotations for knee and brain fully sampled magnetic resonance imaging data. Scientific Data, 9, 152 (2022).

---

> > ### Comment · Reviewer_69mk · 2024-11-22
> >
> > Thank you for including additional metrics, I believe that addresses my question. As to the "physics-heavy" part, yes, updating the title would be most welcome. I guess "model-based" and "scientific" are still too general to me. What I would like to avoid is that publication of this paper would unduly suppress the importance of other inverse problems, that authors didn't cover as "not scientific". For example, certain inverse problems in NLP could be both "scientific" and "model-based", but as you are not covering them they won't be a part of this benchmark. If one comes from a physics background it is tempting to think that all science is physics, but this is not the case. I realise that the argument is more on the pedantic side, but I think it is important as it would have long-lasting consequences for the field. Would authors consider the following names instead "InverseBench: Benchmarking Plug-and-Play Diffusion Models for Physics Inverse Problems" or "InverseBench: Benchmarking Plug-and-Play Diffusion Models for Selected Scientific Inverse Problems"?

---

> > > ### Author Response · Authors · 2024-11-26
> > > **Propose new title**
> > >
> > > We appreciate the reviewer’s thoughtful feedback. It helps us clarify the positioning of our work within the broader landscape of scientific inverse problems. In response, we agree to refine the scope of our paper and propose updating the title to:
> > > “InverseBench: Benchmarking Plug-and-Play Diffusion Models for Inverse Problems in Physical Sciences.”
> > >
> > > We hope this revised title addresses your concerns and look forward to your feedback.

---

### Author Response · Authors · 2024-11-21
**Meta response**

We thank all reviewers for the detailed reviews and constructive feedback that help us further improve our paper. We are glad to find reviewers commenting that our paper is “an important and timely contribution” (69mk), “fills a gap in testing diffusion models” (TH8o), makes “an important contribution to the inverse problems community” (SNSr), and  “contributes a well-curated foundation for other researchers to build upon” (MUia).

---

### Meta-Review · Area_Chair_UYtX · 2024-12-07

**Metareview:**

The paper introduces a benchmark for a series of scientific inverse problems including linear inverse scattering, compressed sensing MRI, and black hole imaging.

All four reviewers recommend acceptance of the paper. The reviewers agree that the paper's contributes to benchmarking diffusion based reconstruction algorithms for a variety of inverse problems, which provides value to the community through standardization and benchmarking of existing methods.

**Additional Comments On Reviewer Discussion:**

The reviewers raise minor issues regarding certain formulations and hyperparameter search that were sorted out during the rebuttal.

---

### Decision · Program_Chairs · 2025-01-22

Accept (Spotlight)